# The m⁶A reader YTHDF2 is a negative regulator for dendrite development and maintenance of retinal ganglion cells

**Fugui Niu[1,2], Peng Han[1], Jian Zhang[1], Yuanchu She[1], Lixin Yang[1], Jun Yu[1], Mengru Zhuang[1], Kezhen Tang[3], Yuwei Shi[1], Baisheng Yang[1], Chunqiao Liu[4], Bo Peng[5,6]\*, Sheng-Jian Ji[1]\***

[1]School of Life Sciences, Department of Biology, Shenzhen Key Laboratory of Gene Regulation and Systems Biology, Brain Research Center, Southern University of Science and Technology, Shenzhen, China; [2]SUSTech-HIT Joint Graduate Program, Southern University of Science and Technology, Shenzhen, China; [3]Shenzhen Institutes of Advanced Technology, Chinese Academy of Sciences, Shenzhen, China; [4]State Key Laboratory of Ophthalmology, Zhongshan Ophthalmic Center, Sun Yat-sen University, Guangzhou, China; [5]Department of Neurosurgery, Jinshan Hospital, Institute for Translational Brain Research, State Key Laboratory of Medical Neurobiology, MOE Frontiers Center for Brain Science, Fudan University, Shanghai, China; [6]Co-Innovation Center of Neuroregeneration, Nantong University, Nantong, China

**\*For correspondence:**
peng@fudan.edu.cn (BP);
jisj@sustech.edu.cn (S-JJ)

**Competing interest:** The authors declare that no competing interests exist.

**Abstract** The precise control of growth and maintenance of the retinal ganglion cell (RGC) dendrite arborization is critical for normal visual functions in mammals. However, the underlying mechanisms remain elusive. Here, we find that the $N^6$-methyladenosine (m⁶A) reader YTHDF2 is highly expressed in the mouse RGCs. Conditional knockout (cKO) of *Ythdf2* in the retina leads to increased RGC dendrite branching, resulting in more synapses in the inner plexiform layer. Interestingly, the *Ythdf2* cKO mice show improved visual acuity compared with control mice. We further demonstrate that *Ythdf2* cKO in the retina protects RGCs from dendrite degeneration caused by the experimental acute glaucoma model. We identify the m⁶A-modified YTHDF2 target transcripts which mediate these effects. This study reveals mechanisms by which YTHDF2 restricts RGC dendrite development and maintenance. YTHDF2 and its target mRNAs might be valuable in developing new treatment approaches for glaucomatous eyes.

## Editor's evaluation

In this study, you propose a role for the m⁶A reader YTHDF2 in regulating the dendritic arbor size of retinal ganglion cells (RGCs). You show that retina-specific loss of *Ythdf2* leads to the expansion of RGC dendritic arbors in the horizontal plane and a widening of the inner plexiform layer (IPL), with *Ythdf2* conditional knockouts showing a modest increase in visual acuity in an optomotor assay. You point to a number of factors downstream of YTHDF2 not previously known to be involved in retinal dendritic development, and propose a role for YTHDF2 in a glaucoma model, in which loss of YTHDF2 is shown to prevent RGC loss. This study presents a careful phenotypic analysis of manipulation of YTHDF2 and provides a foundation for studies on how YTHDF2-mediated mechanisms are integrated into programs of dendritic development and RGC survival.

## Introduction

The mammalian retina is an ideal model system to study neuronal development and neural circuit formation. The retinal ganglion cells (RGCs) are the final and only output neurons in the vertebrate retina and their dendrites collect the electrical information concerning the visual signal from all other cells preceding them. One of the major focuses of research in the retina is to understand how RGC dendrite arborization arises during development (*Prigge and Kay, 2018*). Existing evidences supported that homotypic repulsion controls retinal dendrite patterning (*Lefebvre et al., 2015*). However, in mice which had most RGCs genetically eliminated, the dendrite size and shape of remaining RGCs appeared relatively normal (*Lin et al., 2004*). Thus, the fact that the dendrites of remaining RGCs did not expand to neighboring areas by the remaining RGCs supports the existence of the intrinsic limit for RGC dendrite patterning, which cooperates with the homotypic repulsion to determine the dendrite size of RGCs (*Lefebvre et al., 2015*). However, such intrinsic limiting mechanisms remain elusive.

Glaucoma is one of the leading causes for blindness. The major risk factors for glaucoma include increased intraocular tension. Studies have shown that glaucoma causes pathological changes in RGC dendrites before axon degeneration and soma loss were detected in different model animals (*Weber et al., 1998*; *Shou et al., 2003*; *Morgan et al., 2006*). Thus, elucidation of mechanisms governing RGC dendrite arbor maintenance bears clinical significance.

$N^6$-methyladenosine (m$^6$A) is the most widely distributed and extensively studied internal modification in mRNA (*Dominissini et al., 2012*; *Meyer et al., 2012*; *Nachtergaele and He, 2018*). m$^6$A modification has been shown to regulate brain development and functions in the nervous system (*Livneh et al., 2020*; *Yu et al., 2021a*). By effectors, most of these studies have focused on its demethylases ('m$^6$A erasers') and methyltransferases ('m$^6$A writers'). Since the fate of m$^6$A-modified transcripts is decoded by the m$^6$A-binding proteins ('m$^6$A readers'), how the readers mediate these functions and what are their neural target mRNAs remain to be elucidated. In addition, more precisely controlled spatial-temporal ablation of the m$^6$A readers instead of null knockout is required to elucidate their functions and mechanisms in nervous system.

In this study, we identified an m$^6$A-dependent intrinsic limiting mechanism for RGC dendrite arborization and maintenance. Conditional knockout (cKO) of the m$^6$A reader YTHDF2 in the developing mouse retina increases RGC dendrite branching and improves visual acuity. YTHDF2 also mediates acute ocular hypertension (AOH)-induced RGC degeneration, the experiment model for glaucoma, and *Ythdf2* cKO in the retina alleviates AOH-induced RGC dendrite shrinking and neuronal loss. The regulation of RGC dendrite development and maintenance by YTHDF2 is mediated by two distinct groups of m$^6$A-modified target mRNAs which encode proteins that promote dendrite arborization during development and maintain dendrite tree during injury, respectively. Therefore, our study reveals mechanisms by which YTHDF2 restricts RGC dendrite development and maintenance, which sheds light on developing new treatment approaches for glaucomatous eyes.

## Results

### Knockdown of YTHDF2 leads to a robust increase of RGC dendrite branching

To examine whether m$^6$A modification and its reader proteins play a role in the dendrite development, we utilized the retina as the model system. We first checked their expression patterns in the developing mouse retina. Immunostaining with a widely used m$^6$A antibody demonstrated that RGCs had high m$^6$A modification levels (*Figure 1—figure supplement 1A*). Consistent with the m$^6$A distribution, the m$^6$A reader YTHDF2 is highly expressed in RGCs (*Figure 1A*; *Figure 1—figure supplement 1B*). Conversely, the expression of YTHDF2 in other layers and cells of the retina is much lower or absent (*Figure 1A*; *Figure 1—figure supplement 1B-D*). Another two m$^6$A readers YTHDF1 and YTHDF3 show similar expression patterns (*Figure 1—figure supplement 1E, F*). The strong expression of YTHDFs and high level of m$^6$A modification in RGCs suggest that the m$^6$A reader YTHDFs might play roles in RGC development. We dissected and dissociated the retinal cells and cultured in vitro. We generated lentiviral shRNAs against YTHDFs, which showed similarly efficient knockdown (KD) of YTHDFs in RGC cultures in vitro (*Figure 1B*; *Figure 1—figure supplement 1G,H*). In these YTHDF-deficient RGC cultures, the first and most obvious phenotype that we observed is the robust increase

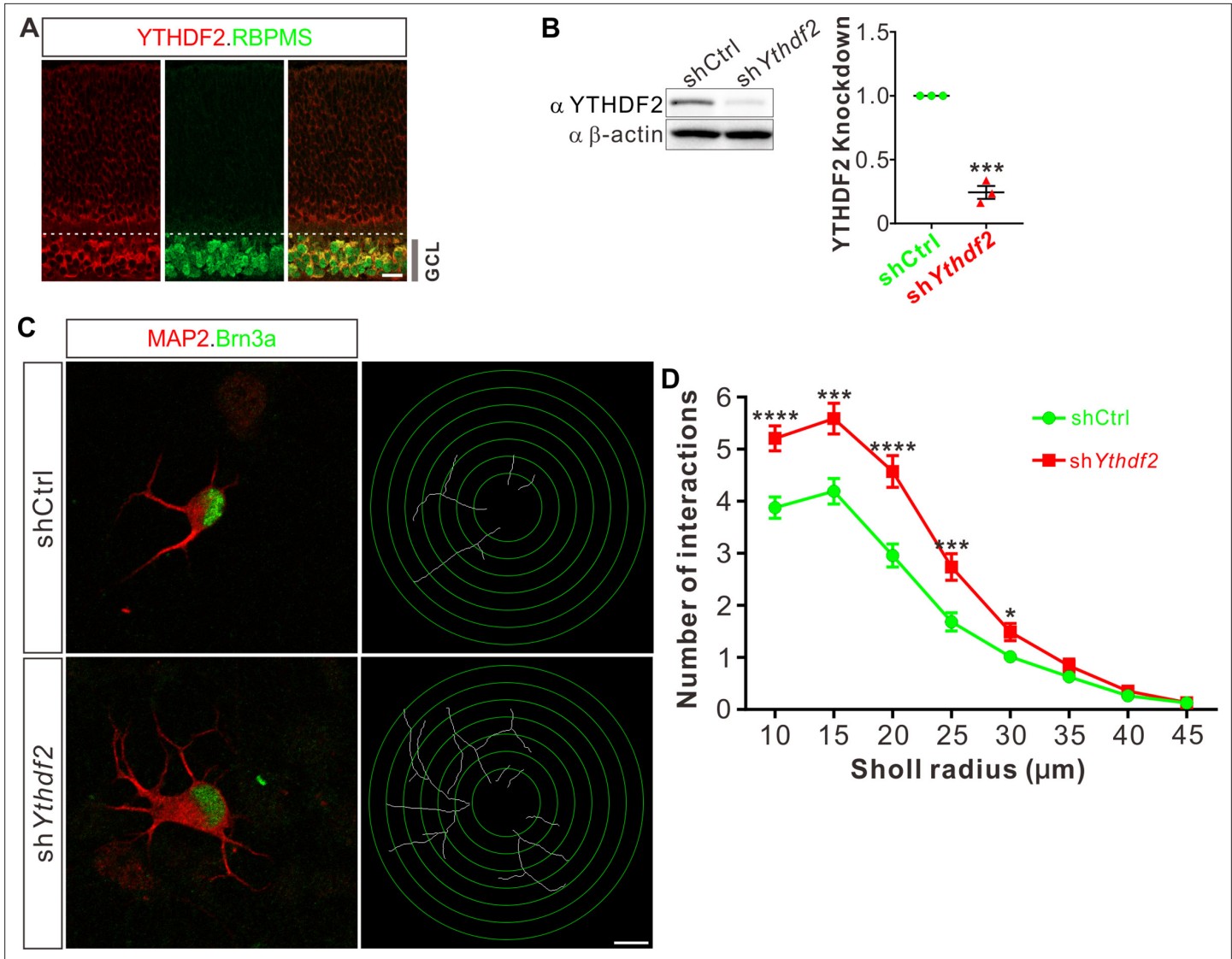

**Figure 1.** Knockdown (KD) of YTHDF2 leads to a robust increase of retinal ganglion cell (RGC) dendrite branching. (**A**) Representative confocal images showing high expression of YTHDF2 in RGCs (marked by RBPMS) in P0 retina. Note that all RGCs marked by the pan-RGC marker RBPMS express YTHDF2 while all YTHDF2-expressing cells are RBPMS⁺ RGCs. GCL, ganglion cell layer. Scale bars: 20 µm. (**B**) Western blotting (WB) confirming efficient KD of YTHDF2 in cultured RGCs using sh*Ythdf2*. Data of WB quantification are mean ± SEM and are represented as dot plots: ***p = 0.00012 (n = 3 replicates); by unpaired Student's *t* test. (**C**) Examination of RGC dendrite development after YTHDF2 KD. As shown, significantly increased branching of dendrites marked by MAP2 immunofluorescence was observed in cultured RGCs marked by Brn3a. Dendrite traces were drawn for the corresponding RGCs. Scale bar: 10 µm. (**D**) Quantification of dendrite branching (**C**) using Sholl analysis. As shown, numbers of interactions are significantly greater in sh*Ythdf2* groups (n = 68 RGCs) than shCtrl groups (n = 72 RGCs) in Sholl radii between 10 and 30 µm. Data are mean ± SEM. ****p = 4.32E-05 (10 µm), ***p = 0.00038 (15 µm), ****p = 2.85E-05 (20 µm), ***p = 0.00084 (25 µm), *p = 0.020 (30 µm), by unpaired Student's *t* test.

The online version of this article includes the following source data and figure supplement(s) for figure 1:

**Source data 1.** Source data for *Figure 1B*.

**Source data 2.** Source data for *Figure 1B*.

**Source data 3.** Source data for *Figure 1B*.

**Figure supplement 1.** Retinal ganglion cells (RGCs) have high level of *N⁶*-methyladenosine (m⁶A) modification and strong expression of YTHDFs.

**Figure supplement 1—source data 1.** Source data for *Figure 1—figure supplement 1E,F*.

**Figure supplement 1—source data 2.** Source data for *Figure 1—figure supplement 1E*.

**Figure supplement 1—source data 3.** Source data for *Figure 1—figure supplement 1E*.

**Figure supplement 1—source data 4.** Source data for *Figure 1—figure supplement 1F*.

**Figure supplement 1—source data 5.** Source data for *Figure 1—figure supplement 1F*.

of dendrite branching of cultured RGCs treated by sh*Ythdf2* (*Figure 1C and D*; *Figure 1—figure supplement 1I,J*). In contrast, the dendrite branching of RGCs with YTHDF1 KD using sh*Ythdf1* was not significantly different from control shRNA (*Figure 1—figure supplement 1K*), while YTHDF3 KD using sh*Ythdf3* caused a slight (statistically significant in several Sholl radii) decrease of RGC dendrite branching compared with control shRNA (*Figure 1—figure supplement 1L*). These results suggest that the m[6]A reader YTHDF2 might play an important role in controlling dendrite branching of RGCs.

## cKO of *Ythdf2* in the retina increases RGC dendrite branching in vivo without disturbing sublaminar targeting

To further explore whether YTHDF2 physiologically regulates RGC dendrite branching in vivo, we generated *Ythdf2* cKO mouse (*Figure 2A*). We used the *Six3-cre* mouse line (*Furuta et al., 2000*), which has been widely used in the field to generate retina-specific knockouts (*Lefebvre et al., 2012*; *Riccomagno et al., 2014*; *Sapkota et al., 2014*; *Krishnaswamy et al., 2015*). YTHDF2 expression is efficiently eliminated in the *Ythdf2* cKO retina compared with their littermate controls at E12.5 (*Figure 2—figure supplement 1A*) and E15.5 (*Figure 2B*). Retina progenitors, amacrine cells, bipolar cells, photoreceptors, horizontal cells, Müller glia, or astrocytes were not affected in *Ythdf2* cKO retina (*Figure 2—figure supplement 1B-K*; *Figure 2—figure supplement 2A-D*), suggesting that YTHDF2 is not involved in the generation or development of these cells. This is in line with the low or no YTHDF2 expression in these cells. The RGC number or density was not affected in the *Ythdf2* cKO retina (*Figure 2C and D*), demonstrating that *Ythdf2* knockout does not disturb RGC neurogenesis. We then cultured RGCs from the *Ythdf2* cKO retina. The dendrite branching of *Ythdf2* cKO RGCs was significantly increased compared with littermate controls (*Figure 2E and F*). RGCs include over 40 subtypes (*Sanes and Masland, 2015*; *Baden et al., 2016*). We thus examined the RGC dendrite branching within different subtypes. One of the RGC subgroups responds preferentially to movement in particular directions and is named the ON-OFF directionally selective RGCs (ooDSGCs). Expression of CART (cocaine- and amphetamine-regulated transcript), a neuropeptide, distinguishes ooDSGCs from other RGCs (*Kay et al., 2011a*). The dendrite branching of ooDSGCs marked by CART/Brn3a co-staining in *Ythdf2* cKO retinal cultures also increased compared with control (*Figure 2G and H*). These data further confirm that the m[6]A reader YTHDF2 regulates dendrite branching of RGCs.

Next, we wanted to confirm this phenotype in vivo by checking specific RGC subtypes. Intravitreal injection of an AAV reporter expressing ZsGreen visualized the dendrite morphology of ooDSGCs marked by CART immunostaining (*Figure 3A*). ooDSGCs showed dramatically increased dendrite branching in *Ythdf2* cKO retina compared with control retina by Sholl analysis (*Figure 3A and B*). The intrinsically photosensitive RGCs (ipRGCs) are unique and melanopsin-expressing cells, which exhibit an intrinsic sensitivity to light (*Hattar et al., 2002*). We analyzed the morphology of ipRGCs visualized by wholemount immunostaining of melanopsin and found that the dendrite branching of ipRGCs was significantly increased in the *Ythdf2* cKO retina (*Figure 3C and D*; *Figure 3—figure supplement 1A-E*). A similar trend was observed in the SMI-32[+]αRGCs (*Figure 3E and F*). These results strongly indicate that the m[6]A reader YTHDF2 negatively regulates RGC dendrite branching in vivo and *Ythdf2* cKO promotes RGC dendrite arborization.

In the retina, RGCs target their dendrites in different sublaminae of the inner plexiform layer (IPL). Since the IPL sublaminar targeting of RGC dendrites is critical for normal visual functions, we wondered whether the increased dendrite branching caused by *Ythdf2* cKO was also accompanied by altered sublaminar patterning of RGC dendrites. We used a *Thy1-GFP* reporter (line O) which labels a few RGCs (*Feng et al., 2000*). As shown in *Figure 3—figure supplement 1F,G*, GFP intensity is generally higher in IPL of the *Ythdf2* cKO retina compared with their littermate controls, which further proves the increased RGC dendrite branching and density. However, the sublaminar pattern of GFP signals looks similar between cKO and littermate control (*Figure 3—figure supplement 1F,G*). Sublaminar dendrite patterning of the ipRGC subtype visualized by immunostaining of melanopsin also demonstrated the similar phenotype (*Figure 3—figure supplement 1H,I*). These data suggest that YTHDF2 has a general control of RGC dendrite branching but has no striking effect on the sublaminar targeting of RGC dendrite. These results are consistent with the previous findings that the RGC dendrite targeting is determined genetically and several transcription factors controlling laminar choice have been identified in RGCs and amacrine cells (*Cherry et al., 2011*; *Kay et al., 2011b*; *Lefebvre et al., 2015*; *Liu et al., 2018*).

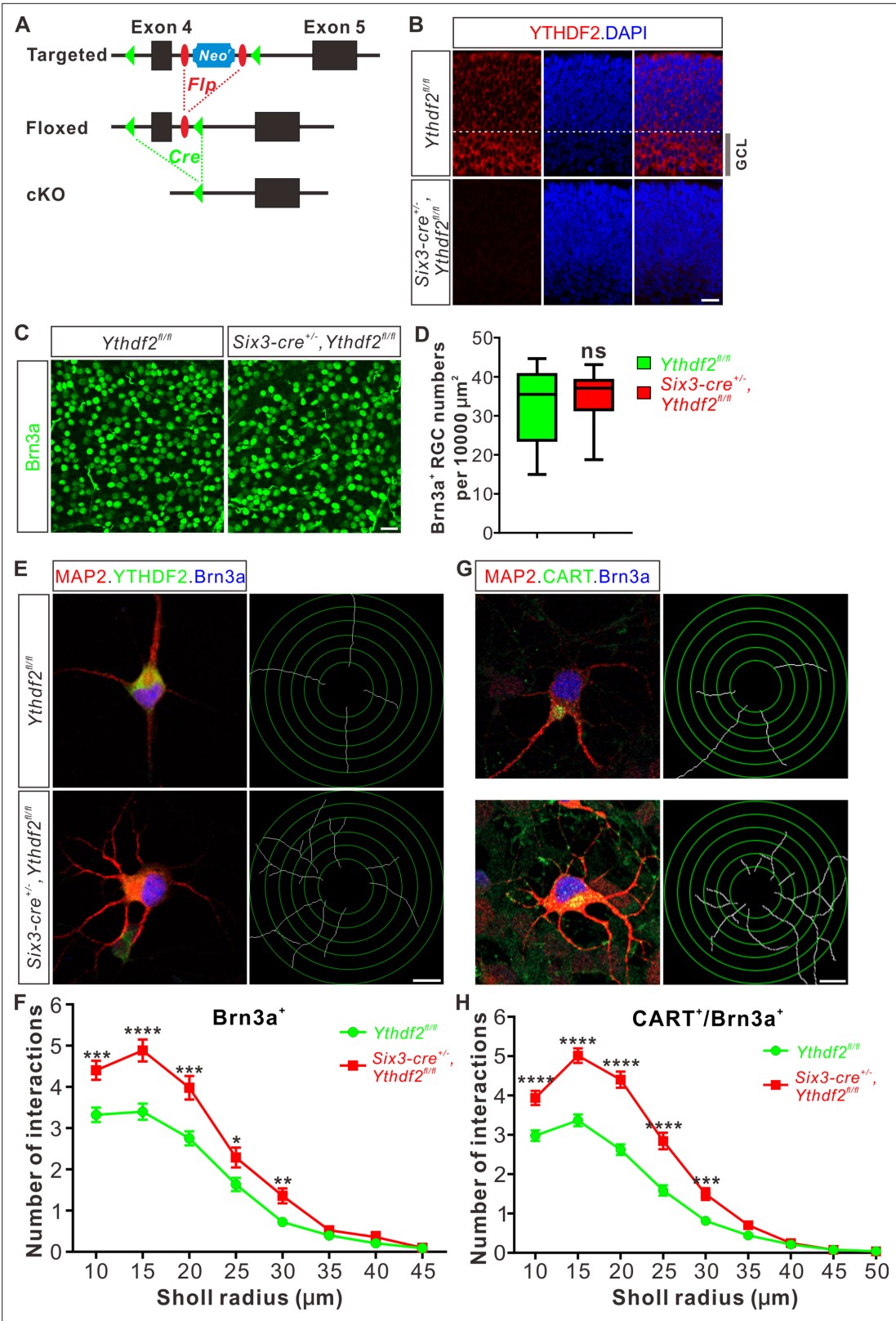

**Figure 2.** Dendrite branching is dramatically increased in cultured retinal ganglion cells (RGCs) from *Ythdf2* conditional knockout (cKO). (**A**) Schematic drawings of the genetic deletion strategy for *Ythdf2*. Exon 4 which contains YTH domain-coding sequence is deleted after Cre-mediated recombination. (**B**) Depletion of YTHDF2 protein in retina of *Six3-cre⁺/⁻;Ythdf2ᶠˡ/ᶠˡ* cKO mice. Anti-YTHDF2 immunostaining of E15.5 retina vertical sections confirmed cKO of YTHDF2 protein, compared with *Ythdf2ᶠˡ/ᶠˡ* littermate controls. Scale bar: 20 µm. (**C, D**) RGC neurogenesis not affected in the *Ythdf2* cKO retina.

*Figure 2 continued on next page*

*Figure 2 continued*

Wholemount immunostaining using a Brn3a antibody was carried out in P20 retina (**C**). Numbers of Brn3a$^+$ RGC per 10,000 µm$^2$ of retina were quantified and showed no difference between the *Ythdf2* cKO and their littermate controls (**D**). n = 12 confocal fields for each genotype. Data are represented as box and whisker plots: ns, not significant (p = 0.79); by unpaired Student's *t* test. Scale bar: 25 µm. (**E**) Examination of RGC dendrite development in *Ythdf2* cKO RGCs. As shown, knockout of YTHDF2 was confirmed by YTHDF2 IF (green). Significantly increased branching of dendrites marked by MAP2 IF (red) was observed in cultured RGCs from the *Ythdf2* cKO retina compared with their littermate controls. Dendrite traces were drawn for the corresponding RGCs. Scale bar: 10 µm. (**F**) Quantification of RGC dendrite branching (**E**) using Sholl analysis. Data are mean ± SEM. Numbers of interactions are significantly greater in *Six3-cre$^{+/-}$,Ythdf2$^{fl/fl}$* groups (n = 68 RGCs) than *Ythdf2$^{fl/fl}$* groups (n = 42 RGCs) in Sholl radii between 10 and 30 µm: ***p = 0.00030 (10 µm), ****p = 1.19E-05 (15 µm), ***p = 0.00018 (20 µm), *p = 0.021 (25 µm), **p = 0.0022 (30 µm), by unpaired Student's *t* test. (**G**) Examination of CART$^+$ (cocaine- and amphetamine-regulated transcript) RGC dendrite development in *Ythdf2* cKO RGCs. Cultured CART$^+$ RGCs from the *Ythdf2* cKO retina have significantly increased branching of dendrites marked by MAP2 IF (red) compared with their littermate controls. Dendrite traces were drawn for the corresponding RGCs. Scale bar: 10 µm. (**H**) Quantification of CART$^+$ RGC dendrite branching (**G**) using Sholl analysis. Data are mean ± SEM. Numbers of interactions are significantly greater in *Six3-cre$^{+/-}$,Ythdf2$^{fl/fl}$* groups (n = 77 RGCs) than *Ythdf2$^{fl/fl}$* groups (n = 90 RGCs) in Sholl radii between 10 and 30 µm: ****p = 3.17E-05 (10 µm), ****p = 6.50E-11 (15 µm), ****p = 5.14E-12 (20 µm), ****p = 5.00E-07 (25 µm), ***p = 0.00020 (30 µm), by unpaired Student's *t* test.

The online version of this article includes the following figure supplement(s) for figure 2:

**Figure supplement 1.** *Ythdf2* conditional knockout (cKO) does not change numbers of retinal progenitors, amacrine cells, bipolar cells, photoreceptors, or horizontal cells.

**Figure supplement 2.** *Ythdf2* conditional knockout (cKO) does not change numbers of Müller glia or astrocytes.

## IPL of *Ythdf2* cKO retina is thicker and has more synapses

The increased dendrite branching of RGCs further prompted us to check whether *Ythdf2* cKO changes IPL development. Immunostaining of P6 retina vertical sections using a MAP2 antibody demonstrated that IPL thickness significantly increased in *Ythdf2* cKO retina (**Figure 4A and B**). As a control, the thicknesses of other retinal layers showed no difference between the *Ythdf2* cKO and control mice (**Figure 4—figure supplement 1A-D**). Quantification of MAP2 IF intensity in IPL suggested that the IPL of *Ythdf2* cKO retina became denser with dendrites (**Figure 4A and C**). These results suggest that the increased dendrite branching results in a thicker and denser IPL in the *Ythdf2* cKO retina.

The IPL of retina is concentrated with synaptic connections, which contain synapses among and between bipolar-amacrine-ganglion cells. The increased RGC dendrite branching and denser IPL in the *Ythdf2* cKO retina prompted us to wonder whether there are changes in synaptic connections in IPL. We used co-staining of the presynaptic marker Bassoon and the postsynaptic marker PSD-95 to count the colocalization puncta of Bassoon$^+$/PSD-95$^+$. We found that the numbers of Bassoon$^+$/PSD-95$^+$ excitatory synapses in IPL of *Ythdf2* cKO retina are significantly larger than that of control retina (**Figure 4D and E**). As a control, the numbers of the excitatory ribbon synapses marked by the colocalization of Bassoon$^+$/PSD-95$^+$ in OPL (outer plexiform layer) show no difference between *Ythdf2* cKO and control retinas (**Figure 4—figure supplement 1E,F**).

All these data verify that the IPL of *Ythdf2* cKO retina is thicker and has more synapses.

## Visual acuity is improved for the *Ythdf2* cKO mice

The features of RGC dendrites, including their size, shape, arborization pattern, and localization, influence the amount and type of synaptic inputs that RGCs receive, which in turn determine how RGCs respond to specific visual stimuli such as the direction of motion (**Liu and Sanes, 2017**). The increased dendrite branching, the thicker and denser IPL, and the more synapses in the IPL inspired us to further explore whether the visual responses of the *Ythdf2* cKO mice were changed or not. *Ythdf2* cKO mice looked normal and had similar body weight and size compared with control mice for either sex (male in **Figure 5A and B**; female in **Figure 5C and D**). The generally normal development of *Ythdf2* cKO mice is consistent with the specific and limited expression of *Six3-cre* in retina (**Figure 5—figure supplement 1A**), and only sparse spots in ventral forebrain (**Figure 5—figure supplement 1B**; **Furuta et al., 2000**). We used an optomotor response (OMR)-based assay (**Prusky et al., 2004**; **Umino et al., 2008**; **Shi et al., 2018**) to monitor visual functions of *Ythdf2* cKO mice (**Figure 5E**). Surprisingly, the *Ythdf2* cKO mice showed modestly improved visual acuity compared with the control mice, measuring spatial frequency threshold as 0.45 ± 0.0043 c/deg (cycle per degree) and 0.43 ± 0.0085 c/deg, respectively (**Figure 5F**, male mice). Similar phenotype was observed in female mice (**Figure 5G**). These results suggest that the visual acuity is modestly improved in the *Ythdf2* cKO mice.

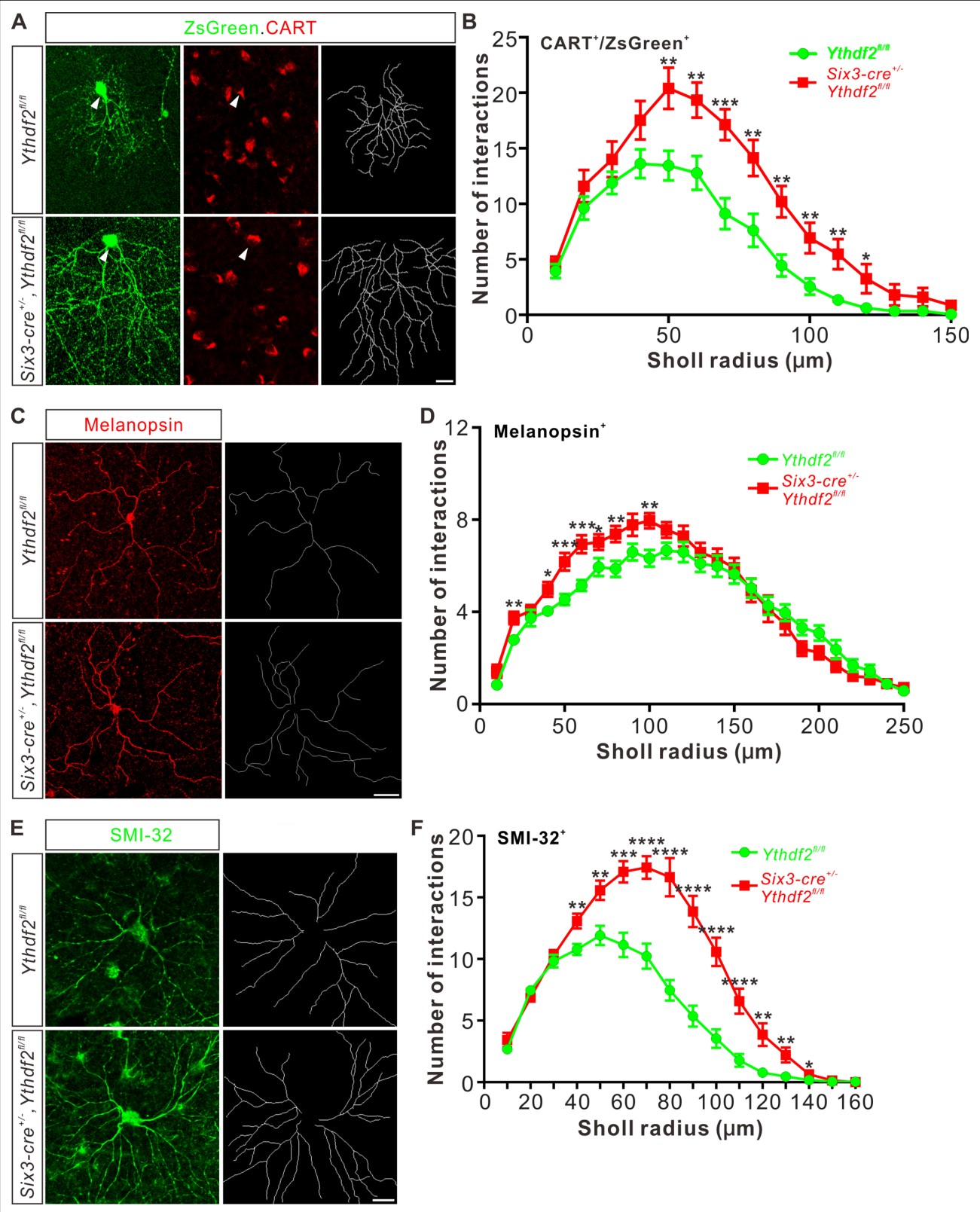

**Figure 3.** Dendrite branching of specific retinal ganglion cell (RGC) subtypes increases in *Ythdf2* conditional knockout (cKO) in vivo. (**A**) Co-labeling of ON-OFF directionally selective RGCs (ooDSGCs) by AAV-ZsGreen and CART (cocaine- and amphetamine-regulated transcript) IF in vivo. Intravitreal injection of AAV-expressing ZsGreen reporter was performed at P17 and retinas were collected at P27. The white arrowheads indicate ooDSGCs co-labeled by ZsGreen and CART IF, which show dramatically increased dendrite branching in *Ythdf2* cKO compared with control. Dendrite traces were

*Figure 3 continued on next page*

Figure 3 continued

drawn for the corresponding RGCs shown. Scale bar: 20 μm. (**B**) Quantification of dendrite branching of ZsGreen⁺/CART⁺ ooDSGCs (**A**) using Sholl analysis. Data are mean ± SEM. Numbers of interactions are significantly greater in *Six3-cre⁺/⁻,Ythdf2^fl/fl* groups (n = 15 RGCs) than *Ythdf2^fl/fl* groups (n = 18 RGCs) in Sholl radii between 50 and 120 μm: **p = 0.0041 (50 μm), **p = 0.0059 (60 μm), ***p = 0.00036 (70 μm), **p = 0.0058 (80 μm), **p = 0.0018 (90 μm), **p = 0.0064 (100 μm), **p = 0.0045 (110 μm), *p = 0.040 (120 μm), by unpaired Student's *t* test. (**C**) Dendrites of intrinsically photosensitive RGCs (ipRGCs) visualized by wholemount immunostaining of P20 retina using a melanopsin antibody in vivo. Dendrite traces were drawn for the corresponding RGCs shown. Scale bar: 50 μm. (**D**) Quantification of dendrite branching of melanopsin⁺ ipRGCs (**C**) using Sholl analysis. Data are mean ± SEM. Numbers of interactions are significantly greater in *Six3-cre⁺/⁻,Ythdf2^fl/fl* groups (n = 18 RGCs) than *Ythdf2^fl/fl* groups (n = 21 RGCs) in Sholl radii between 20 and 100 μm: **p = 0.0083 (20 μm), *p = 0.018 (40 μm), ***p = 0.00068 (50 μm), ***p = 0.00027 (60 μm), *p = 0.048 (70 μm), **p = 0.0048 (80 μm), **p = 0.0023 (100 μm), by unpaired Student's *t* test. (**E**) Dendrites of αRGCs visualized by wholemount immunostaining of P20 retina using an SMI-32 antibody in vivo. Dendrite traces were drawn for the corresponding RGCs shown. Scale bar: 20 μm. (**F**) Quantification of dendrite branching of SMI-32⁺αRGCs (**E**) using Sholl analysis. Data are mean ± SEM. Numbers of interactions are significantly greater in *Six3-cre⁺/⁻,Ythdf2^fl/fl* groups (n = 14 RGCs) than *Ythdf2^fl/fl* groups (n = 22 RGCs) in Sholl radii between 40 and 140 μm: **p = 0.0044 (40 μm), **p = 0.0035 (50 μm), ***p = 0.00021 (60 μm), ****p = 2.63E-05 (70 μm), ****p = 2.38E-06 (80 μm), ****p = 1.68E-06 (90 μm), ****p = 6.76E-06 (100 μm), ****p = 5.72E-05 (110 μm), **p = 0.0011 (120 μm), **p = 0.0032 (130 μm), *p = 0.047 (140 μm), by unpaired Student's *t* test.

The online version of this article includes the following figure supplement(s) for figure 3:

**Figure supplement 1.** General dendrite density in inner plexiform layer (IPL) is increased without affecting sublaminar targeting.

This phenotype is most likely attributed to the increased RGC dendrite branching and thicker and denser IPL with more synapses because all other parts and processes of retina are not affected except RGC dendrite in the *Ythdf2* cKO mediated by *Six3-cre* (*Figure 2—figure supplement 1* and *Figure 4—figure supplement 1*). The eyes and optic fibers also showed no difference between *Ythdf2* cKO and control mice (*Figure 5—figure supplement 1C-E*). We further checked the targeting of optic nerves to the brain by anterograde labeling with cholera toxin subunit B (CTB) and found no difference of retinogeniculate or retinocollicular projections between *Ythdf2* cKO and control mice (*Figure 5—figure supplement 1F, G*), suggesting the guidance and central targeting of RGC axons are not affected in the *Ythdf2* cKO.

## YTHDF2 target mRNA were identified with transcriptomic and proteomic analysis

Next, we continued to explore the underlying molecular mechanisms of the effects on dendrite branching caused by *Ythdf2* cKO in the retina. First, we wanted to know what transcripts YTHDF2 recognizes and binds. We carried out anti-YTHDF2 RNA immunoprecipitation (RIP) in the retina followed by RNA sequencing of the elute (RIP-Seq). Two biological replicates of anti-YTHDF2 RIP-Seq identified 1638 transcripts (*Supplementary file 1*). Functional annotation of YTHDF2 RIP targets revealed significant enrichment in cellular component terms such as neuron part and neuron projection, and biological process terms such as cellular component organization and neuron projection development. We further zoomed in to check neural terms in cellular component (*Figure 6A*) and biological process (*Figure 6B*). We found that substantial numbers of YTHDF2 target transcripts are involved in cytoskeleton, dendrite, and their organization and development (*Figure 6A and B*), which is consistent with the dendrite branching phenotype observed in the *Ythdf2* cKO retina.

The working model for YTHDF2 is that it binds and destabilizes its m⁶A-modified target transcripts (*Wang et al., 2014*). Since the destabilization of mRNAs will eventually decrease their protein levels, we carried out proteome analysis using mass spectrometry (MS) in acute sh*Ythdf2*-mediated KD of cultured RGCs, in order to identify directly affected targets. Three biological replicates of YTHDF2 KD followed by MS (YTHDF2 KD/MS) identified 114 proteins which were upregulated by YTHDF2 KD (*Supplementary file 2*). Functional annotation of these proteins revealed significant enrichment in neuron development- and cytoskeleton-related terms (*Figure 6C*), which is similar to anti-YTHDF2 RIP-Seq results.

By overlapping the two gene lists screened from anti-YTHDF2 RIP-Seq (*Supplementary file 1*) and YTHDF2 KD/MS_upregulation (*Supplementary file 2*), we identified a group of potential YTHDF2 target mRNAs in RGCs (*Supplementary file 3*), including *Kalrn*, *Strn*, and *Ubr4*. m⁶A modification of these mRNAs was verified by anti-m⁶A pulldown (*Figure 6D*). *Kalrn* (*Kalirin*) gene generates three alternative splicing isoforms *Kalrn7*, *Kalrn9*, and *Kalrn12* encoding guanine-nucleotide exchange factors for Rho GTPases, which have been shown to regulate hippocampal and cortical dendritic

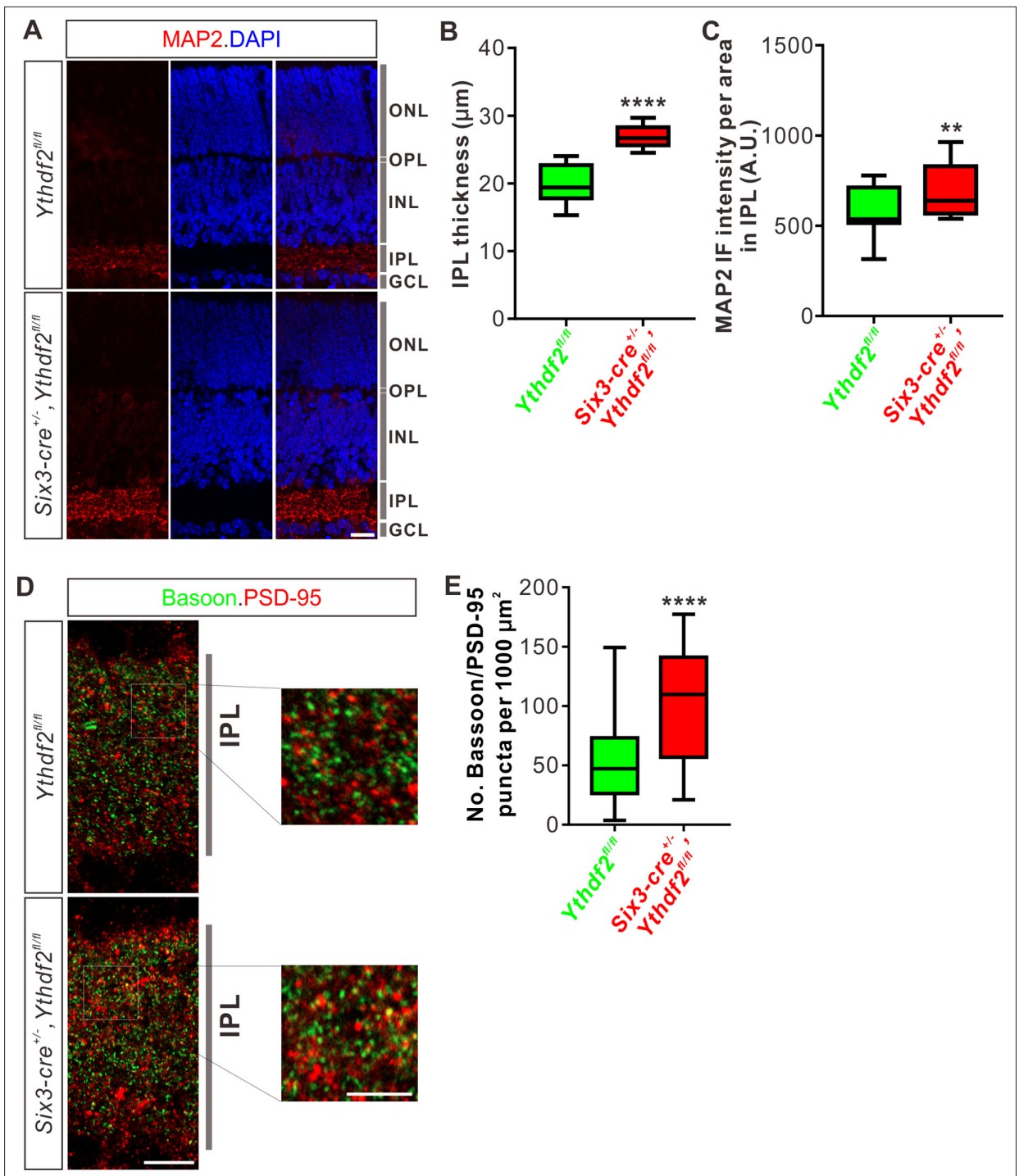

**Figure 4.** Inner plexiform layer (IPL) of the *Ythdf2* conditional knockout (cKO) retina is thicker and has more synapses. (**A**) Cross-sections of P6 *Six3-cre*<sup>+/-</sup> *,Ythdf2*<sup>fl/fl</sup> retina showing increased IPL thickness by MAP2 staining compared with littermate control. ONL, outer nuclear layer; OPL, outer plexiform layer; INL, inner nuclear layer; IPL, inner plexiform layer; GCL, granule cell layer. Scale bar: 20 μm. (**B, C**) Quantification showing increased IPL thickness and MAP2 IF intensity per area in IPL of the *Ythdf2* cKO retina (**A**). Quantification data are represented as box and whisker plots: ****p = 1.28E-07 for **B** (n = 12 sections for each genotype), by unpaired Student's *t* test; **p = 0.0045 for **C** (n = 12 sections for each genotype), by paired Student's *t* test. (**D, E**) Representative confocal images showing the excitatory synapses labeled by colocalization of Basoon (presynaptic) and PSD-95 (postsynaptic) in the IPL of P30 retina (**D**). There are significantly more synapses in the *Ythdf2* cKO IPL compared with control. Quantification data are represented as box and whisker plots (**E**): n = 47 confocal fields for *Ythdf2*<sup>fl/fl</sup>, n = 23 confocal fields for *Six3-cre*<sup>+/-</sup>,*Ythdf2*<sup>fl/fl</sup>; ****p = 1.63E-05; by unpaired Student's *t* test. Scale bars: 10 μm (**D**) and 5 μm (inset in **D**).

*Figure 4 continued on next page*

*Figure 4 continued*

The online version of this article includes the following figure supplement(s) for figure 4:

**Figure supplement 1.** Thickness or synapse numbers in outer plexiform layer (OPL) shows no difference between the *Ythdf2* conditional knockout (cKO) and control retinas.

branching (*Xie et al., 2010*; *Yan et al., 2015*), and are required for normal brain functions (*Penzes et al., 2001*; *Xie et al., 2007*; *Cahill et al., 2009*; *Russell et al., 2014*; *Lu et al., 2015*; *Herring and Nicoll, 2016*). Strn (Striatin) was first identified in striatum, and functions as a B subunit of the serine/threonine phosphatase PP2A and is also a core component of a multiprotein complex called STRIPAK (striatin-interacting phosphatase and kinase complex) (*Benoist et al., 2006*; *Li et al., 2018*). Strn was reported to regulate dendritic arborization only in striatal neurons but not in cortical neurons (*Li et al., 2018*). However, whether and how Kalrn and Strn work in the retina was still unknown. Ubr4 (ubiquitin protein ligase E3 component N-recognin 4) is also known as p600 and has been shown to play roles in neurogenesis, neuronal migration, neuronal signaling, and survival (*Parsons et al., 2015*). However, whether Ubr4 regulates dendrite development remains elusive.

## YTHDF2 controls the stability of its target mRNAs which encode proteins regulating RGC dendrite branching

MS analysis after YTHDF2 KD has shown that the protein levels of these target mRNAs were upregulated (*Supplementary file 2*). IF using antibodies against Strn and Ubr4 detected specific signals in the IPL which were increased in *Ythdf2* cKO retina compared with control retina (*Figure 7—figure supplement 1A*). Enrichment of these proteins in IPL implies that these proteins might function locally in RGC dendrites to regulate dendrite development.

We next wanted to know whether YTHDF2 controlled the protein levels of these m$^6$A-modified target mRNAs through regulation of translation or transcript stability. As shown in *Figure 7—figure supplement 1B-E*, the mRNA levels of *Kalrn7*, *Kalrn9*, *Kalrn12*, *Strn,* and *Ubr4* were dramatically increased after KD of YTHDF2, KO of *Ythdf2*, or KD of METTL14, supporting that YTHDF2 might regulate stability of these target mRNAs. We further evaluated potential changes in the stability of these target mRNAs in an m$^6$A-dependent manner. We further verified this by directly measuring the stability of these target mRNAs. As shown in *Figure 7A*, all the target mRNAs showed significantly increased stability in the *Ythdf2* cKO retina compared with controls. These results suggest that YTHDF2 controlled the protein levels of its m$^6$A-modifed target mRNAs by decreasing their stability.

Next we explored the functions of these YTHDF2 target mRNAs in RGC dendrite development. We first generated siRNAs against these transcripts (*Figure 7—figure supplement 1F*). We then checked the effects on RGC dendrite branching after KD of these target mRNAs by siRNAs in cultured RGCs. As shown in *Figure 7B*, KD of *Kalrn7*, *Kalrn9*, *Kalrn12*, *Strn,* or *Ubr4* led to significant decreases of RGC dendrite branching. Interestingly, the siCocktail against all these target mRNAs further significantly reduced the RGC dendrite branching compared with each individual siRNA (*Figure 7—figure supplement 1G*), suggesting that these targets may work in different pathways to regulate the RGC dendrite morphology. We further examined whether these target mRNAs mediate YTHDF2-regulated RGC dendrite branching. As shown in *Figures 2E–H , and 3*, cKO of *Ythdf2* led to increased dendrite branching of RGCs both in vitro and in vivo. Transfection of siRNAs against these target mRNAs rescued dendrite branching increases in cultured *Ythdf2* cKO RGCs (*Figure 7C*). We continued to generate and performed intravitreal injection of AAV viral sh*Kalrn12* and sh*Ubr4*, which significantly rescued dendrite branching increases of CART$^+$ ooDSGCs and SMI-32$^+$αRGCs in *Ythdf2* cKO retina in vivo (*Figure 7D*).

Taken together, we identified a group of YTHDF2 target mRNAs that encode proteins regulating RGC dendrite branching, which mediate YTHDF2-controlled RGC dendrite branching.

## *Ythdf2* cKO retina is more resistant to AOH

The glaucomatous eyes are symptomatized with progressive neurodegeneration and vision loss (*Agostinone and Di Polo, 2015*). High intraocular pressure is a major risk factor in glaucoma and has been shown to cause pathological changes in RGC dendrites before axon degeneration or soma loss is detected in different model animals (*Weber et al., 1998*; *Shou et al., 2003*; *Morgan et al.,*

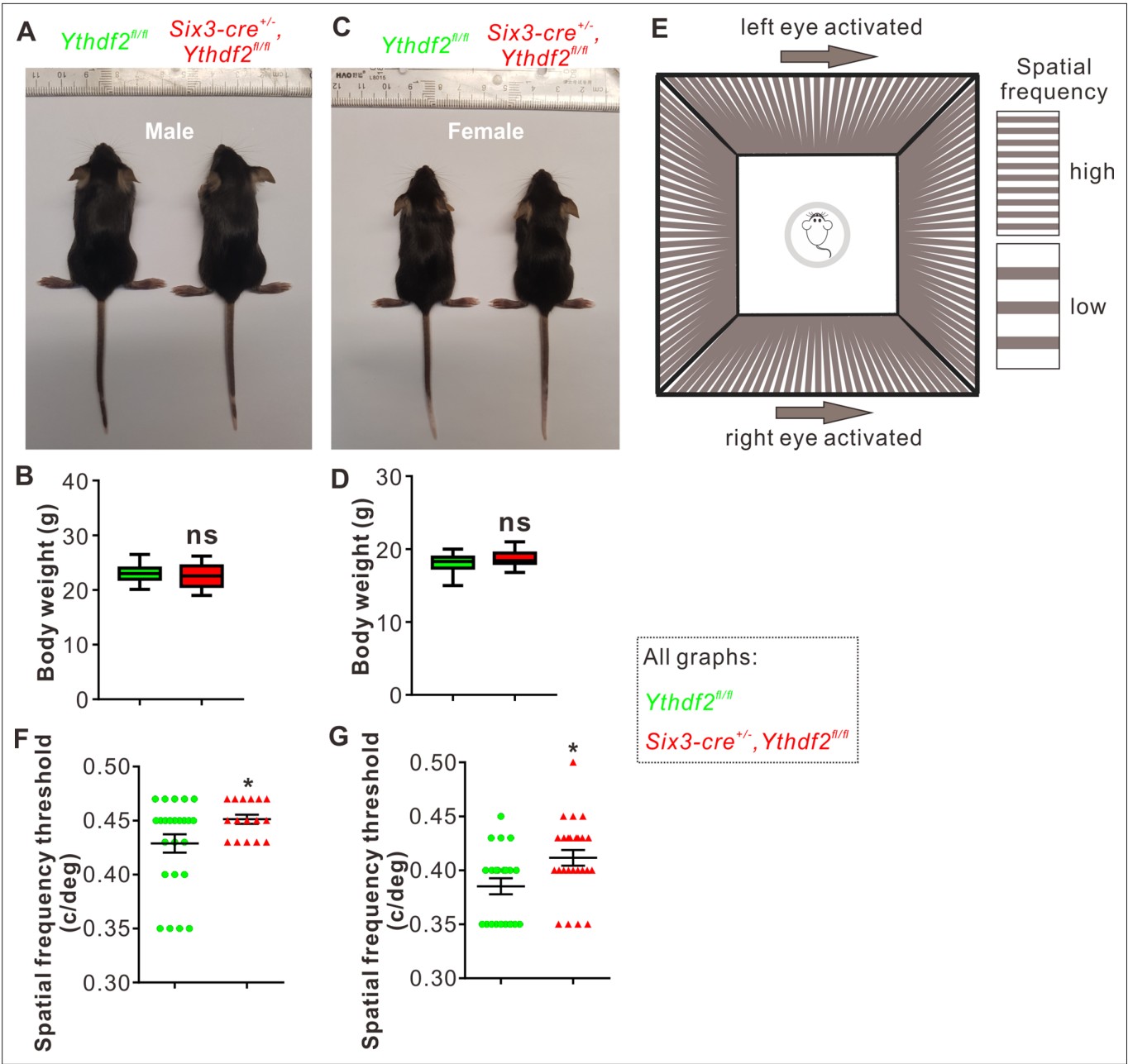

**Figure 5.** Visual acuity is improved for the *Ythdf2* conditional knockout (cKO) mice. (**A–D**) Six3-Cre-mediated *Ythdf2* cKO showing normal animal development and body weight (male in **A**, female in **C**). Quantification data of body weight (**B, D**) are represented as box and whisker plots: p = 0.41 in **B** (male, n = 24 for control, n = 18 for cKO); *P* = 0.08 in **D** (female, n = 23 for control, n = 25 for cKO); ns, not significant; by unpaired Student's *t* test. (**E**) The setup of optomotor response assay is illustrated by schematic drawing. (**F, G**) Optomotor response assay demonstrating improved visual acuity in the *Ythdf2* cKO mice. Quantification data are mean ± SEM: *p = 0.048 in **F** (male, n = 24 control, n = 16 cKO); *p = 0.015 in **G** (female, n = 21 control, n = 25 cKO); by unpaired Student's *t* test.

The online version of this article includes the following figure supplement(s) for figure 5:

**Figure supplement 1.** Guidance or central targeting of optic nerves is not affected in Six-Cre-mediated *Ythdf2* conditional knockout (cKO).

2006). Our findings that *Ythdf2* cKO in retina promotes RGC dendrite branching during development inspired us to wonder whether YTHDF2 also regulates RGC dendrite maintenance in the acute glaucoma model caused by AOH. We utilized the AOH model made with control and *Ythdf2* cKO mice to check whether *Ythdf2* cKO in the retina could alter the pathology in the glaucomatous eyes. RGC dendrite branching is significantly decreased after AOH operation compared with non-AOH in

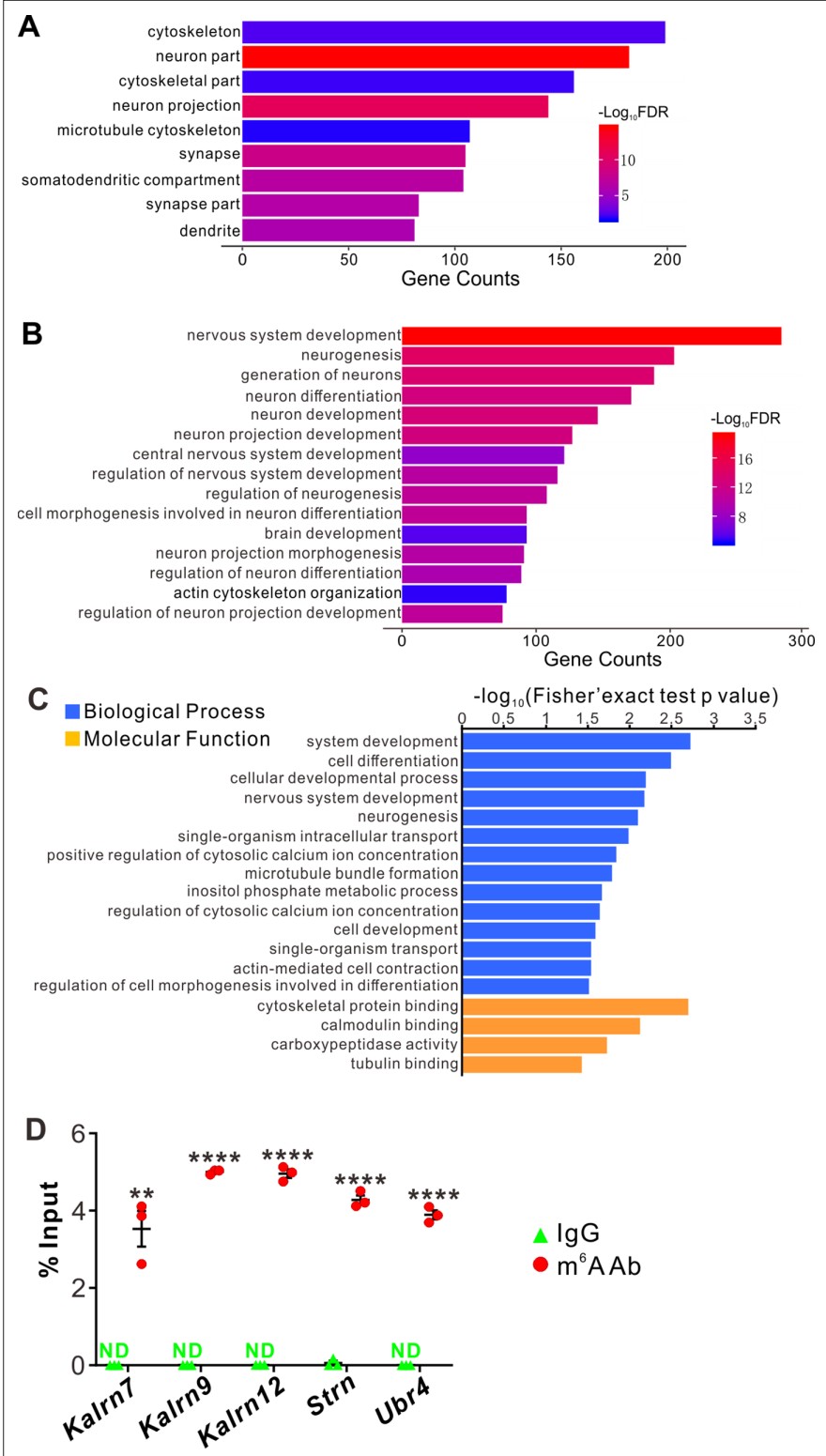

**Figure 6.** YTHDF2 target mRNAs were identified with transcriptomic and proteomic analysis. (**A, B**) Gene Ontology (GO) analysis of YTHDF2 target transcripts identified by anti-YTHDF2 RNA immunoprecipitation (RIP) in the retina followed by RNA sequencing (RIP-Seq). Neural terms were picked out in cellular component (**A**) and biological process (**B**). (**C**) GO analysis of proteins which are upregulated after YTHDF2 knockodwn (KD) by mass spectrometry (MS). (**D**) Verification of $N^6$-methyladenosine (m$^6$A) modification of YTHDF2 target mRNAs by anti-

*Figure 6 continued on next page*

*Figure 6 continued*

m⁶A pulldown followed by RT-qPCR. ND, not detected. Data are mean ± SEM and are represented as dot plots (n = 3 replicates): **p = 0.0016 for *Kalrn7*; ****p = 1.40E-08 for *Kalrn9*; ****p = 1.46E-06 for *Kalrn12*; ****p = 5.46E-06 for *Strn*; ****p = 4.90E-06 for *Ubr4*; by unpaired Student's *t* test.

either genotype (*Figure 8—figure supplement 1A,B*). Interestingly, the *Ythdf2* cKO retina with AOH operation maintains significantly higher dendrite complexity compared with the glaucomatous eyes of *Ythdf2^fl/fl^* control mice (*Figure 8A and B*). In addition, there are significant RGC neuron losses in both genotypes after AOH (*Figure 8C and D*). However, the reduction of RGC number in the *Ythdf2* cKO retina is less than control retina (*Figure 8C and D*). These results support that *Ythdf2* cKO protects retina from RGC dendrite degeneration and soma loss caused by AOH.

Next we wanted to know whether and how YTHDF2 target mRNAs mediate these effects in the AOH models. We first checked the expression of YTHDF2 target mRNAs identified in the developing retina (*Supplementary file 3*) in the adult *Ythdf2* cKO and control retina. We found that two target mRNAs *Hspa12a* and *Islr2* show upregulation in the adult *Ythdf2* cKO retina compared with control (*Figure 8—figure supplement 1C*). m⁶A modification of *Hspa12a* and *Islr2* mRNAs was further verified by anti-m⁶A pulldown (*Figure 8—figure supplement 1D*). *Hspa12a* encodes heat shock protein A12A which is an atypical member of the heat shock protein 70 family and has been shown to be downregulated in diseases such as ischemic stroke, schizophrenia, and renal cell carcinoma (*Pongrac et al., 2004*; *Mao et al., 2018*; *Min et al., 2020*). *Islr2* encodes immunoglobulin superfamily containing leucine-rich repeat protein two and is poorly studied. Here, we found that *Hspa12a* and *Islr2* are downregulated in the retina after AOH operation (*Figure 8—figure supplement 1E*), which is likely caused by upregulation of YTHDF2 in the AOH-treated retina (*Figure 8—figure supplement 1F-H*). We therefore hypothesized that AOH upregulates YTHDF2 which in turn downregulates its targets *Hspa12a* and *Islr2*, thus causing RGC dendrite degeneration and soma loss. If this is the case, overexpression of *Hspa12a* and *Islr2* might protect RGC dendrite from AOH-triggered degeneration. We thus generated AAV harboring overexpression constructs of *Hspa12a* and *Islr2* which were intravitreally injected to wild type retinas. After the AOH induction, the retinas overexpressing *Hspa12a* and *Islr2* maintain significantly more complex RGC dendrite arbor and show better RGC survival compared with control AAV (*Figure 8E–G*).

These data verify that loss-of-function of YTHDF2 and gain-of-function of its targets *Hspa12a* and *Islr2* have neuroprotective roles in the glaucomatous retina.

## Discussion

Functions and mechanisms of mRNA m⁶A modification in the dendrite development were not known. Here, we revealed a critical role of the m⁶A reader YTHDF2 in RGC dendrite development and maintenance. YTHDF2 have two phases of function to control RGC dendrite development first and then maintenance through regulating two sets of target mRNAs. In early postnatal stages, the target mRNAs *Kalrn7*, *Kalrn9*, *Kalrn12*, *Strn,* and *Ubr4* mediate YTHDF2 functions to regulate RGC dendrite development. In the adult mice, another set of target mRNAs *Hspa12a* and *Islr2* mediate YTHDF2 function to regulate RGC dendrite maintenance.

### Positive and negative regulators for dendrite development

The general principle for dendrite arborization is that the dendrite arbor cannot be either too big or too small in order to precisely sample a presynaptic target area during neural circuit formation (*Lefebvre et al., 2015*). Numerous extrinsic and intrinsic mechanisms have been found to regulate dendritic arbor patterning, which involves both positive and negative factors to achieve balanced control of dendritic growth (*Jan and Jan, 2010*; *Dong et al., 2015*; *Ledda and Paratcha, 2017*). For the secreted and diffusible cues, BDNF promotes dendrite branching and complexity (*Cheung et al., 2007*); the non-canonical Wnt7b/PCP pathway is a positive regulator of dendrite growth and branching (*Rosso et al., 2005*); the non-canonical Wnt receptor Ryk works as a negative regulator by limiting the extent of dendritic branching (*Lanoue et al., 2017*). For the contact-mediated signals, the cadherins Celsr2 and Celsr3 regulate dendrite growth in an opposite manner in cortical pyramidal and Purkinje neurons, and hippocampal neurons, respectively (*Shima et al., 2004*; *Shima et al., 2007*). For

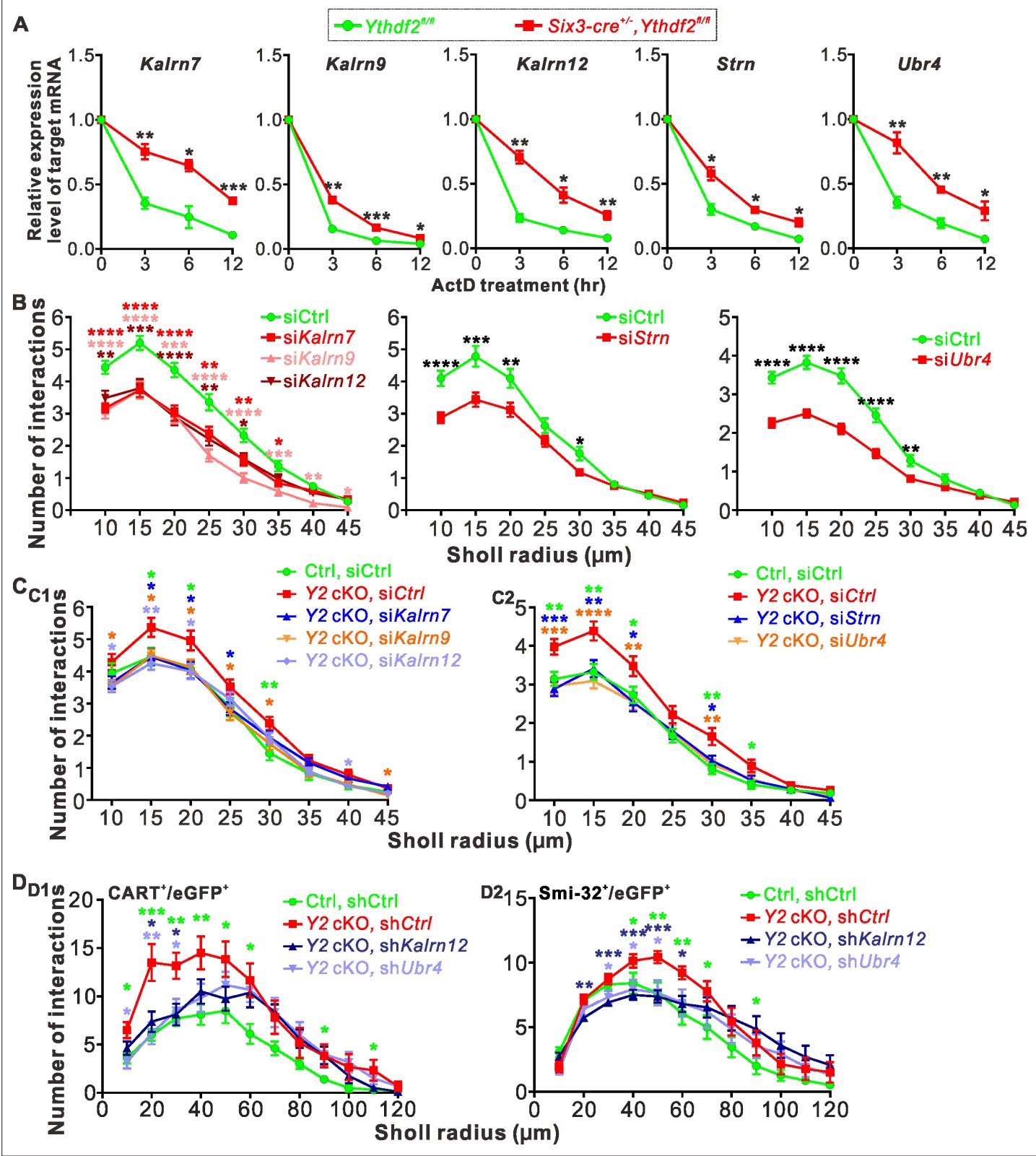

**Figure 7.** YTHDF2 target mRNAs mediate YTHDF2-controlled retinal ganglion cell (RGC) dendrite branching. (**A**) YTHDF2 target mRNAs showing increased stability in the *Ythdf2* conditional knockout (cKO) retina. RGCs dissected from E14.5 *Ythdf2* cKO and control embryos were cultured, treated with actinomycin D (ActD), and collected at different timepoints. Data are mean ± SEM (n = 3 replicates). For *Kalrn7*, **p = 0.0057 (3 hr), *p = 0.014 (6 hr), ***p = 0.00039 (12 hr); for *Kalrn9*, **p = 0.0036 (3 hr), ***p = 0.00090 (6 hr), *p = 0.032 (12 hr); for *Kalrn12*, **p = 0.0012 (3 hr), *p = 0.010 (6 hr), **p

*Figure 7 continued on next page*

*Figure 7 continued*

= 0.0069 (12 hr); for *Strn*, *p = 0.014 (3 hr), *p = 0.012 (6 hr), *p = 0.016 (12 hr); for *Ubr4*, **p = 0.0077 (3 hr), **p = 0.0059 (6 hr), *p = 0.041 (12 hr); all by unpaired Student's *t* test. (**B**) Knockdown (KD) of the target mRNAs causing decreased dendrite branching of cultured RGCs prepared from wild type (WT) E14.5 retina by Sholl analysis. Brn3a and MAP2 IF were used to mark RGCs and visualize dendrites. Data are mean ± SEM. For *Kalrn7* (n = 59 for siCtrl, n = 56 for si*Kalrn7*), ****p = 2.33E-06 (10 μm), ****p = 5.85E-06 (15 μm), ****p = 8.67E-05 (20 μm), **p = 0.0045 (25 μm), **p = 0.0058 (30 μm), *p = 0.010 (35 μm); for *Kalrn9* (n = 59 for siCtrl, n = 46 for si*Kalrn9*), ****p = 3.69E-05 (10 μm), ****p = 5.53E-05 (15 μm), ***p = 0.00020 (20 μm), ****p = 3.09E-06 (25 μm), ****p = 4.63E-06 (30 μm), ***p = 0.00059 (35 μm), **p = 0.0010 (40 μm), *p = 0.042 (45 μm); for *Kalrn12* (n = 59 for siCtrl, n = 39 for si*Kalrn12*), **p = 0.0031 (10 μm), ***p = 0.00017 (15 μm), ****p = 6.56E-05 (20 μm), **p = 0.0017 (25 μm), *p = 0.017 (30 μm); for *Strn* (n = 51 for siCtrl, n = 57 for si*Strn*), ****p = 4.19E-05 (10 μm), **p = 0.00067 (15 μm), **p = 0.0079 (20 μm), *p = 0.015 (30 μm); for *Ubr4* (n = 81 for siCtrl, n = 81 for si*Ubr4*), ****p = 1.26E-08 (10 μm), ****p = 7.61E-10 (15 μm), ****p = 2.35E-08 (20 μm), ****p = 1.39E-05 (25 μm), **p = 0.0061 (30 μm); all by unpaired Student's *t* test. (**C**) Increased dendrite branching of cultured RGCs prepared from E14.5 *Ythdf2* cKO (*Y2* cKO) retina was rescued by KD of target mRNAs using siRNAs. Data are mean ± SEM. Ctrl, *Ythdf2*^fl/fl; *Y2* cKO, *Six3-cre*^+/-,*Ythdf2*^fl/fl. In **C1**, '*Ctrl*, siCtrl' (n = 35 neurons) vs. '*Y2* cKO, siCtrl' (n = 52 neurons), *p = 0.038 (15 μm), *p = 0.045 (20 μm), **p = 0.0036 (30 μm); '*Y2* cKO, si*Kalrn7*' (n = 55 neurons) vs. '*Y2* cKO, siCtrl', *p = 0.020 (15 μm), *p = 0.025 (20 μm), *p = 0.031 (25 μm); '*Y2* cKO, si*Kalrn9*' (n = 66 neurons) vs. '*Y2* cKO, siCtrl', *p = 0.020 (10 μm), *p = 0.013 (15 μm), *p = 0.031 (20 μm), *p = 0.017 (25 μm), *p = 0.031 (30 μm), *p = 0.031 (45 μm); '*Y2* cKO, si*Kalrn12*' (n = 80 neurons) vs. '*Y2* cKO, siCtrl', *p = 0.015 (10 μm), **p = 0.0018 (15 μm), *p = 0.015 (20 μm), *p = 0.027 (40 μm). In **C2**, '*Ctrl*, siCtrl' (n = 50 neurons) vs. '*Y2* cKO, siCtrl' (n = 47 neurons), **p = 0.0031 (10 μm), **p = 0.0013 (15 μm), *p = 0.029 (20 μm), **p = 0.0015 (30 μm), *p = 0.014 (35 μm); '*Y2* cKO, si*Strn*' (n = 45 neurons) vs. '*Y2* cKO, siCtrl', ***p = 0.00016 (10 μm), **p = 0.0043 (15 μm), *p = 0.010 (20 μm), *p = 0.018 (30 μm); '*Y2* cKO, si*Ubr4*' (n = 57 neurons) vs. '*Y2* cKO, siCtrl', ***p = 0.00084 (10 μm), ****p = 4.89E-05 (15 μm), **p = 0.0058 (20 μm), **p = 0.0045 (30 μm). All by unpaired Student's *t* test. (**D**) Increased dendrite branching of RGC subtypes in *Ythdf2* cKO (*Y2* cKO) retina was rescued by KD of target mRNAs through intravitreal injection of AAV shRNAs in vivo. Data are mean ± SEM. Ctrl, *Ythdf2*^fl/fl; *Y2* cKO, *Six3-cre*^+/-,*Ythdf2*^fl/fl. In **D1** (CART^+/eGFP^+ ooDSGCs), '*Ctrl*, shCtrl' (n = 10 neurons) vs. '*Y2* cKO, shCtrl' (n = 6 neurons), *p = 0.010 (10 μm), ***p = 0.00049 (20 μm), **p = 0.0021 (30 μm), **p = 0.0047 (40 μm), *p = 0.028 (50 μm), *p = 0.011 (60 μm), *p = 0.030 (90 μm), *p = 0.042 (110 μm); '*Y2* cKO, sh*Kalrn12*' (n = 8 neurons) vs. '*Y2* cKO, shCtrl', *p = 0.012 (20 μm), *p = 0.014 (30 μm); '*Y2* cKO, sh*Ubr4*' (n = 6 neurons) vs. '*Y2* cKO, shCtrl', *p = 0.011 (10 μm), **p = 0.0084 (20 μm), *p = 0.029 (30 μm). In **D2** (SMI-32^+αRGCs), '*Ctrl*, shCtrl' (n = 14 neurons) vs. '*Y2* cKO, shCtrl' (n = 14 neurons), *p = 0.032 (40 μm), **p = 0.0019 (50 μm), **p = 0.0014 (60 μm), *p = 0.015 (70 μm), *p = 0.044 (90 μm); '*Y2* cKO, sh*Kalrn12*' (n = 26 neurons) vs. '*Y2* cKO, shCtrl', **p = 0.0023 (20 μm), ***p = 0.00076 (30 μm), ***p = 0.00030 (40 μm), ***p = 0.00020 (50 μm), *p = 0.015 (60 μm); '*Y2* cKO, sh*Ubr4*' (n = 15 neurons) vs. '*Y2* cKO, shCtrl', *p = 0.042 (30 μm), *p = 0.024 (40 μm), *p = 0.018 (50 μm). All by unpaired Student's *t* test.

The online version of this article includes the following figure supplement(s) for figure 7:

**Figure supplement 1.** YTHDF2 target mRNAs were characterized and validated.

the transcription factors, studies have shown that manipulation of Cux1 and Cux2 levels has distinct effects on apical and basal arbors of cortical dendrites (*Cubelos et al., 2015*); interestingly, the functions of Sp4 in dendrite development are dependent on the cellular context of its expression, for example, Sp4 promotes dendrite growth and branching in hippocampal dentate granule cells but limits dendrite branching in cerebellar granule cells (*Ramos et al., 2007*; *Zhou et al., 2007*). Here, we identified another negative regulator YTHDF2 which works posttranscriptionally, and loss-of-function of YTHDF2 increased dendrite complexity during development and protected RGC degeneration from AOH. We have validated this effect on several RGC subtypes. Since there are dozens of RGC subtypes, it is technically challenging but still interesting to test whether this effect is universal to all subtypes or there is specificity for RGC subtypes.

## Posttranscriptional regulation of dendrite development

It is well established that mRNAs can be transported and targeted to specific neuronal compartments such as axons and dendrites. Local translation of these mRNAs enables exquisite and rapid control of local proteome in specific subcellular compartments (*Ledda and Paratcha, 2017*). Local translation is known to play roles in controlling dendrite arborization (*Chihara et al., 2007*), and is regulated by specific RNA-binding proteins (*Jan and Jan, 2010*). In *Drosophila*, the RNA-binding proteins Pumilio (Pum), Nanos (Nos), Glorund (Glo), and Smaug (Smg) regulate morphogenesis and branching of specific classes of dendritic arborization neurons through controlling translation of their target mRNAs including *nanos* mRNA itself (*Ye et al., 2004*; *Brechbiel and Gavis, 2008*). The mouse homologue of another RNA-binding protein Staufen, Stau1, regulates dendritic targeting of ribonucleoprotein particles and dendrite branching (*Vessey et al., 2008*). Here, we found that the m⁶A reader and RNA-binding protein YTHDF2 control stability of its target mRNAs and regulate dendrite branching in RGCs. It would be interesting to see whether these target mRNAs are localized into dendrites and whether YTHDF2 works in dendrites to control their stability and translation. Actually, *Strn4* mRNA has been shown to be present in dendrites and locally translated (*Lin et al., 2017*). In addition, how the

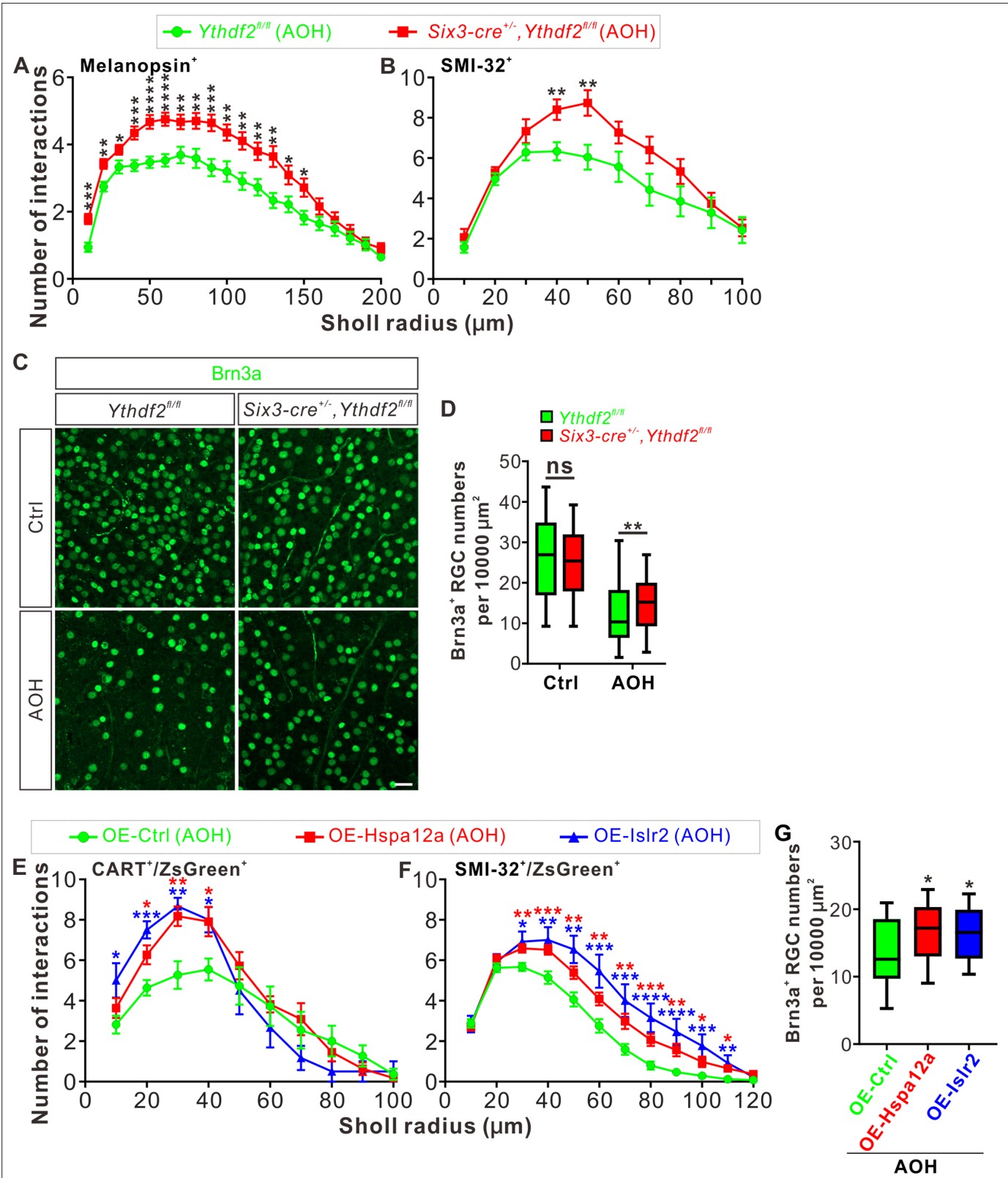

**Figure 8.** *Ythdf2* conditional knockout (cKO) retina is more resistant to acute ocular hypertension (AOH). (**A, B**) Better maintenance of retinal ganglion cell (RGC) dendrite arborization in *Ythdf2* cKO retina after AOH operation. AOH was performed using adult mice, and retinas were collected after AOH for wholemount immunostaining of melanopsin and SMI-32 to visualize the dendrite arbors of corresponding RGC subtype, respectively. Dendrite traces were drawn as previously shown and quantification of dendrite branching was done using Sholl analysis. Data are mean ± SEM. Numbers

*Figure 8 continued on next page*

*Figure 8 continued*

of interactions are significantly greater in *Six3-cre*$^{+/-}$,*Ythdf2*$^{fl/fl}$ retina than *Ythdf2*$^{fl/fl}$ control retina in both RGC subtypes after AOH: for melanopsin$^+$ intrinsically photosensitive RGCs (ipRGCs) in **A**, *Ythdf2*$^{fl/fl}$/AOH (n = 51 RGCs) vs. cKO/AOH (n = 64 RGCs), ***p = 0.00015 (10 µm), **p = 0.0017 (20 µm), *p = 0.034 (30 µm), ***p = 0.00035 (40 µm), ****p = 3.02E-05 (50 µm), ****p = 2.63E-05 (60 µm), **p = 0.0029 (70 µm), **p = 0.0028 (80 µm), ***p = 0.00035 (90 µm), **p = 0.0032 (100 µm), **p = 0.0014 (110 µm), **p = 0.0043 (120 µm), **p = 0.0014 (130 µm), *p = 0.023 (140 µm), *p = 0.013 (150 µm); for SMI-32$^+$αRGCs in **B**, *Ythdf2*$^{fl/fl}$/AOH (n = 21 neurons) vs. cKO/AOH (n = 15 neurons), **p = 0.0052 (40 µm), **p = 0.0057 (50 µm); all by unpaired Student's *t* test. (**C, D**) *Ythdf2* cKO retina showing less severe RGC loss after AOH. AOH was performed using adult mice and retinas were collected after AOH for wholemount immunostaining using a Brn3a antibody (**C**). Numbers of Brn3a$^+$ RGCs per 10,000 µm$^2$ of retina were quantified for different genotypes and conditions (confocal fields for analysis: n = 117 for *Ythdf2*$^{fl/fl}$/Ctrl; n = 98 for *Ythdf2*$^{fl/fl}$/AOH; n = 110 for cKO/Ctrl; n = 104 for cKO/AOH). Data are represented as box and whisker plots (**D**): ns, not significant (p = 0.16; *Ythdf2*$^{fl/fl}$/Ctrl vs. cKO/Ctrl); **p = 0.0077 (*Ythdf2*$^{fl/fl}$/AOH vs. cKO/AOH); by unpaired Student's *t* test. Scale bar: 25 µm. (**E, F**) Overexpression (OE) of YTHDF2 targets *Hspa12a* and *Islr2* protecting retina from RGC dendrite degeneration in the AOH model. Wild type (WT) mice were intravitreally injected with AAV overexpressing *Hspa12a* or *Islr2* and then operated with AOH. Wholemount immunostaining of CART/ZsGreen and SMI-32/ZsGreen was carried out to visualize the dendrite arbors of corresponding RGC subtype, respectively. Dendrite traces were drawn as previously shown and quantification of dendrite branching was done using Sholl analysis. Data are mean ± SEM. Numbers of interactions are significantly greater in retina with OE of *Hspa12a* or *Islr2* than control retina in both RGC subtypes after AOH. For CART$^+$ ON-OFF directionally selective RGCs (ooDSGCs) in **E**: OE-Ctrl/AOH (n = 11 RGCs) vs. OE-Hspa12a/AOH (n = 11 RGCs), *p = 0.014 (20 µm), **p = 0.0025 (30 µm), *p = 0.018 (40 µm); OE-Ctrl/AOH vs. OE-Islr2/AOH (n = 6 RGCs), *p = 0.024 (10 µm), ***p = 0.00031 (20 µm), **p = 0.0038 (30 µm), *p = 0.013 (40 µm). For SMI-32$^+$αRGCs in **F**: OE-Ctrl/AOH (n = 49 neurons) vs. OE-Hspa12a/AOH (n = 46 neurons), **p = 0.0023 (30 µm), ***p = 0.00080 (40 µm), **p = 0.0059 (50 µm), **p = 0.0051 (60 µm), **p = 0.0036 (70 µm), ***p = 0.00070 (80 µm), **p = 0.0015 (90 µm), *p = 0.016 (100 µm), *p = 0.011 (110 µm); OE-Ctrl/AOH vs. OE-Islr2/AOH (n = 13 RGCs), *p = 0.010 (30 µm), **p = 0.0093 (40 µm), **p = 0.0019 (50 µm), ***p = 0.00085 (60 µm), ***p = 0.00067 (70 µm), ****p = 4.25E-05 (80 µm), ****p = 2.54E-05 (90 µm), ***p = 0.00020 (100 µm) , **p = 0.0016 (110 µm). All by unpaired Student's *t* test. (**G**) OE of YTHDF2 targets *Hspa12a* and *Islr2* alleviating RGC loss in the AOH model. WT mice were intravitreally injected with AAV overexpressing *Hspa12a* or *Islr2* and then operated with AOH. Wholemount immunostaining of Brn3a was performed to label RGCs. Numbers of Brn3a$^+$ RGCs per 10,000 µm$^2$ of retina were quantified for different conditions (confocal fields for analysis: n = 19 for OE-Ctrl; n = 26 for OE-Hspa12a; n = 24 for OE-Islr2). Data are represented as box and whisker plots: *p = 0.034 (OE-Hspa12a vs. OE-Ctrl; *p = 0.029 (OE-Islr2 vs. OE-Ctrl); by unpaired Student's *t* test.

The online version of this article includes the following figure supplement(s) for figure 8:

**Figure supplement 1.** *Hspa12a* and *Islr2* are two target mRNAs of YTHDF2 in adult retina.

proteins encoded by these target mRNAs regulate RGC dendrite branching during development and maintenance remains to be explored and will be important future directions.

## Neuroprotective genes in retinal injuries and degeneration

Transcriptome analyses have revealed differentially expressed genes after retinal injuries such as AOH-induced glaucoma and optic nerve crush (ONC), and the upregulated genes are of importance for discovering new treatment approaches (*Jakobs, 2014*; *Tran et al., 2019*). One of the previous studies has identified *Mettl3*, encoding the m$^6$A writer, as an upregulated gene after ONC (*Agudo et al., 2008*). Here, we found *Ythdf2*, encoding an m$^6$A reader, was also upregulated in the retina after AOH. We further found that *Hspa12a* and *Islr2*, two targets of YTHDF2 in adult retina, were downregulated in glaucomatous retinas. Overexpression of *Hspa12a* and *Islr2* protected retina from AOH-caused RGC dendrite degeneration. Our findings in this study suggest that YTHDF2 and its neuroprotective target mRNAs might be valuable in developing novel therapeutic approaches to treat neurodegeneration caused by glaucoma and other retinal injuries.

## Materials and methods

### Key resources table

| Reagent type (species) or resource | Designation | Source or reference | Identifiers | Additional information |
|---|---|---|---|---|
| Strain, strain background (mouse) | Mouse: Ythdf2$^{fl/fl}$ | *Yu et al., 2021b* | N/A | |
| Strain, strain background (mouse) | Mouse: Tg(Six3-cre)69Frty/GcoJ | Jackson Laboratory | Cat#: JAX_019755 RRID: IMSR_JAX:019755 | |
| Strain, strain background (mouse) | Mouse: B6.Cg-Tg(Thy1-EGFP)OJrs/GfngJ | Jackson Laboratory | Cat#: JAX_007919 RRID: IMSR_JAX:007919 | |
| Strain, strain background (mouse) | Mouse: B6.129 × 1-Gt(ROSA)26Sor$^{tm1(EYFP)Cos}$/J | Jackson Laboratory | Cat#: JAX_006148 RRID: IMSR_JAX:006148 | |

*Continued on next page*

*Continued*

| Reagent type (species) or resource | Designation | Source or reference | Identifiers | Additional information |
|---|---|---|---|---|
| Antibody | Anti-GFP (Chicken polyclonal) | Abcam | Cat#: ab13970, RRID: AB_300798 | IF (1:1000) |
| Antibody | Anti-MAP2 (Chicken polyclonal) | Abcam | Cat#: ab5392, RRID: AB_2138153 | IF (1:10,000) |
| Antibody | Anti-RBPMS (Guinea pig polyclonal) | PhosphoSolutions | Cat#: 1832-RBPMS, RRID: AB_2492226 | IF (1:1000) |
| Antibody | Anti-VAChT (Goat polyclonal) | Millipore | Cat#: ABN100, RRID: AB_2630394 | IF (1:1000) |
| Antibody | Anti-β Actin (Mouse monoclonal) | Abcam | Cat#: ab6276, RRID: AB_2223210 | WB (1:30,000) |
| Antibody | Anti-β Actin (Mouse monoclonal) | ABclonal | Cat#: AC004, RRID: AB_2737399 | WB (1:30,000) |
| Antibody | Anti-AP2α (Mouse monoclonal) | DSHB | Cat#: 3B5, RRID: AB_2313947 | IF (1:1000) |
| Antibody | Anti-Bassoon (Mouse monoclonal) | Enzo Life Sciences | Cat#: ADI-VAM-PS003, RRID: AB_10618753 | IF (1:2500) |
| Antibody | Anti-Brn3a (Mouse monoclonal) | Millipore | Cat#: MAB1585, RRID: AB_94166 | IF (1:300) |
| Antibody | Anti-Calbindin-D-28K (Mouse monoclonal) | Sigma-Aldrich | Cat#: C9848, RRID: AB_476894 | IF (1:200) |
| Antibody | Anti-PKCα (Mouse monoclonal) | Santa Cruz Biotechnology | Cat#: sc-8393, RRID: AB_628142 | IF (1:500) |
| Antibody | Anti-SMI-32 (Mouse monoclonal) | BioLegend | Cat#: 801701, RRID: AB_2564642 | IF (1:200) |
| Antibody | Anti-Strn (Striatin) (Mouse monoclonal) | BD Biosciences | Cat#: 610838, RRID: AB_398157 | IF (1:500) |
| Antibody | Anti-CART (Rabbit polyclonal) | Phoenix Pharmaceuticals | Cat#: H-003–62, RRID: AB_2313614 | IF (1:2000) |
| Antibody | Anti m$^6$A (Rabbit polyclonal) | Synaptic Systems | Cat# 202003, RRID: AB_2279214 | IF (1:200) |
| Antibody | Anti-melanopsin (Rabbit polyclonal) | Thermo Fisher Scientific | Cat#: PA1-780, RRID: AB_2267547 | IF (1:1000) |
| Antibody | Anti-PKCα (Rabbit polyclonal) | Cell Signaling | Cat#: CST-2056 | IF (1:1000) |
| Antibody | Anti-PSD-95 (Mouse monoclonal) | Abcam | Cat#: ab2723, RRID: AB_303248 | IF (1:500) |
| Antibody | Anti-Recoverin (Rabbit polyclonal) | Millipore | Cat#: AB5585, RRID: AB_2253622 | IF (1:1000) |
| Antibody | Anti-YTHDF2 (Rabbit polyclonal) | Proteintech | Cat#: 24744–1-AP, RRID: AB_2687435 | IF (1:1000) |
| Antibody | Anti-YTHDF1 (Rabbit polyclonal) | Proteintech | Cat#: 17479–1-AP, RRID: AB_2217473 | IF (1:1000) |
| Antibody | Anti-YTHDF3 (Rabbit polyclonal) | Abcam | Cat#: ab103328, RRID: AB_10710895 | IF (1:1000) |
| Antibody | Anti-Ubr4 (Rabbit polyclonal) | Abcam | Cat#: ab86738, RRID: AB_1952666 | IF (1:300) |
| Antibody | Anti-Chx10 (Sheep polyclonal) | Exalpha | Cat#: X1179P | IF (1:1000) |
| Antibody | Anti-GFAP (Chicken polyclonal) | Millipore | Cat#: AB5541, RRID: AB_177521 | IF (1:500) |
| Antibody | Anti-Lhx2 (Goat polyclonal) | Santa Cruz Biotechnology | Cat#: sc-19344, RRID: AB_2135660 | IF (1:200) |

*Continued on next page*

*Continued*

| Reagent type (species) or resource | Designation | Source or reference | Identifiers | Additional information |
|---|---|---|---|---|
| Antibody | Anti-Lhx2 (Rabbit monoclonal) | Abcam | Cat#: ab184337 | IF (1:500) |
| Antibody | Anti-chicken IgY (Alexa 488 donkey) | Jackson Immunoresearch | Cat#: 703-545-155, RRID: AB_2340375 | IF (1:500) |
| Antibody | Anti-G. pig IgG (Alexa 488 donkey) | Jackson Immunoresearch | Cat#: 706-545-148, RRID: AB_2340472 | IF (1:500) |
| Antibody | Anti-mouse IgG (Alexa 488 donkey) | Thermo Fisher Scientific | Cat#: A-21202, RRID: AB_141607 | IF (1:500) |
| Antibody | Anti-rabbit IgG (Alexa 488 donkey) | Thermo Fisher Scientific | Cat#: A-21206, RRID: AB_141708 | IF (1:500) |
| Antibody | Anti-goat IgG (Alexa 555 donkey) | Thermo Fisher Scientific | Cat#: A-21432, RRID: AB_2535853 | IF (1:1000) |
| Antibody | Anti-mouse IgG (Alexa 555 donkey) | Thermo Fisher Scientific | Cat#: A-31570, RRID: AB_2536180 | IF (1:1000) |
| Antibody | Anti-rabbit IgG (Alexa 555 donkey) | Thermo Fisher Scientific | Cat#: A-31572, RRID: AB_162543 | IF (1:1000) |
| Antibody | Anti-sheep IgG (Alexa 555 donkey) | Thermo Fisher Scientific | Cat#: A-21436, RRID: AB_2535857 | IF (1:1000) |
| Antibody | Anti-chicken IgY (Alexa 555 goat) | Thermo Fisher Scientific | Cat#: A-21437, RRID: AB_2535858 | IF (1:1000) |
| Antibody | Anti-mouse IgG (Alexa 647 donkey) | Thermo Fisher Scientific | Cat#: A-31571, RRID: AB_162542 | IF (1:200) |
| Antibody | Anti-mouse IgG (HRP donkey) | Abcam | Cat#: ab97030, RRID: AB_10680919 | WB (1:2500) |
| Antibody | Anti-rabbit IgG (HRP donkey) | Abcam | Cat#: ab16284, RRID: AB_955387 | WB (1:2500) |
| Antibody | Anti-mouse IgG (HRP VHH) | AlpaLife | Cat#: KTSM1321 | WB (1:5000) |
| Antibody | Anti-rabbit IgG (HRP VHH) | AlpaLife | Cat#: KTSM1322 | WB (1:5000) |
| Recombinant DNA reagent | Plasmid: pLKO.1-TRC | Addgene | Addgene plasmid #10878, RRID: Addgene_10878 | |
| Sequence-based reagent | shRNA targeting sequence of negative control | This paper | N/A | GCATCAAGGTG AACTTCAAGA |
| Sequence-based reagent | shRNA targeting sequence of mouse *Ythdf2* | *Yu et al., 2018* | N/A | GGACGTTCCC AATAGCCAACT |
| Sequence-based reagent | shRNA targeting sequence of mouse *Ythdf1* | This paper | N/A | GGACATTGGT ACTTGGGATAA |
| Sequence-based reagent | shRNA targeting sequence of mouse *Ythdf3* | This paper | N/A | GGATTTGGCAA TGATACTTTG |
| Sequence-based reagent | shRNA targeting sequence of mouse *Mettl14#6* | This paper | N/A | GCTGGACCTGG GATGATATTA |
| Sequence-based reagent | shRNA targeting sequence of mouse *Mettl14#7* | This paper | N/A | CCCAGCTTGT ACTTTGCTTTA |
| Sequence-based reagent | shRNA targeting sequence of negative control (AAV) | This paper | N/A | TTCTCCGAAC GTGTCACGTAA |
| Sequence-based reagent | shRNA targeting sequence of mouse *Kalrn12* | This paper | N/A | TGATGAGCTGA TGGAAGAA |
| Sequence-based reagent | shRNA targeting sequence of mouse *Ubr4* | This paper | N/A | AATGATGAGC AGTCATCTC |
| Sequence-based reagent | siRNA targeting sequence of negative control | *Yu et al., 2018* | N/A | UUCUCCGAAC GUGUCACGUTT |

*Continued on next page*

*Continued*

| Reagent type (species) or resource | Designation | Source or reference | Identifiers | Additional information |
|---|---|---|---|---|
| Sequence-based reagent | siRNA targeting sequence of mouse *Kalrn7* | *Xie et al., 2007* | N/A | AGUACAAUCCU GGCCAUGUTT |
| Sequence-based reagent | siRNA targeting sequence of mouse *Kalrn9* | *Yan et al., 2015* | N/A | ACUGGACUGG ACUUCUAUUTT |
| Sequence-based reagent | siRNA targeting sequence of mouse *Kalrn12* | *Yan et al., 2015* | N/A | CGAUGAGCUG AUGGAAGAATT |
| Sequence-based reagent | siRNA targeting sequence of mouse *Strn* | *Breitman et al., 2008* | N/A | GGUGAAGAUCG AGAUACAATT |
| Sequence-based reagent | siRNA targeting sequence of mouse *Ubr4* | *Shim et al., 2008* | N/A | AAUGAUGAGC AGUCAUCUATT |
| Sequence-based reagent | qPCR primers of mouse *18*s | *Wang et al., 2018* | N/A | Fwd: GCTTAATTTGACT CAACACGGGA Rev: AGCTATCAATCTG TCAATCCTGTC |
| Sequence-based reagent | qPCR primers of mouse *Gapdh* | *Mains et al., 2011* | N/A | Fwd: TTGTCAGCAATG CATCCTGCACCACC Rev: CTGAGTGGCAGT GATGGCATGGAC |
| Sequence-based reagent | qPCR primers of mouse *Ythdf2* | This paper | N/A | Fwd: GAGCAGAGA CCAAAAGGTCAAG Rev: CTGTGGGCTC AAGTAAGGTTC |
| Sequence-based reagent | qPCR primers of mouse *Kalrn7* | *Mains et al., 2011* | N/A | Fwd: GATACCATATCCAT TGCCTCCAGGACC Rev: CCAGGCTGCGC GCTAAACGTAAG |
| Sequence-based reagent | qPCR primers of mouse *Kalrn9* | *Mains et al., 2011* | N/A | Fwd: GCCCCTCGCC AAAGCCACAGC Rev: CCAGTGAGT CCCGTGGTGGGC |
| Sequence-based reagent | qPCR primers of mouse *Kalrn12* | *Mains et al., 2011* | N/A | Fwd: CAGCAGCCA CGTGCCTGCAGC Rev: TCTTGACATTGGG AATGGGCCGCAC |
| Sequence-based reagent | qPCR primers of mouse *Strn* | This paper | N/A | Fwd: TGAAGCCTG GAATGTGGACC Rev: CTATTGGGC CTCTTCACCCC |
| Sequence-based reagent | qPCR primers of mouse *Ubr4* | This paper | N/A | Fwd: TGAGTGAGG ACAAGGGCAAC Rev: GGGTTGGAT CGAACGAAGGT |
| Sequence-based reagent | qPCR primer for mouse *Hspa12a* | This paper | N/A | Fwd: GGGTTTGCACA GGCTAAGGA Rev: TCTGATGGACG GTCAGGTCT |
| Sequence-based reagent | qPCR primer for mouse *Islr2* | This paper | N/A | Fwd: GAAGCTCCCTTA GACTGTCACC Rev: CCCCATCGTGA CTCCTGCTG |
| Sequence-based reagent | PCR primer for mouse *Hspa12a* CDS | This paper | N/A | Fwd: ATGGCGGACAA GGAAGCTGG Rev: GTAATTTAAGAA GTCGATCCCC |
| Sequence-based reagent | PCR primer for mouse *Islr2* CDS | This paper | N/A | Fwd: ATGGGGCC CTTTGGAGC Rev: GCCCGCTGTC TGCCTGTAG |

*Continued on next page*

*Continued*

| Reagent type (species) or resource | Designation | Source or reference | Identifiers | Additional information |
|---|---|---|---|---|
| Sequence-based reagent | Mouse genotyping primers for *Ythdf2* loxp site 1 | This paper | N/A | GCTTGTAGTTATG TTGTGTACCAC and GCAGCTCTGACT ATTCTAAAACCTCC |
| Sequence-based reagent | Mouse genotyping primers for *Ythdf2* loxp site 2 | This paper | N/A | CTCATAACATCC ATAGCCACAGG and CCAAGAGATAG CTTTCCTAATG |
| Sequence-based reagent | Mouse genotyping primers for *Six3-cre* | Chunqiao Liu's lab | N/A | CCTTCCTCCCT CTCTATGTG and GAACGAACCT GGTCGAAATC |
| Sequence-based reagent | Mouse genotyping primers for *Thy1-GFP* | The Jackson Laboratory website | N/A | CGGTGGTGC AGATGAACTT and ACAGACACAC ACCCAGGACA |
| Sequence-based reagent | Mouse genotyping primers for Rosa-YFP mutant site | The Jackson Laboratory website | N/A | AGGGCGAGG AGCTGTTCA and TGAAGTCGAT GCCCTTCAG |
| Sequence-based reagent | Mouse genotyping primers for Rosa-YFP wild type site | The Jackson Laboratory website | N/A | CTGGCTTCT GAGGACCG and CAGGACAAC GCCCACACA |
| Peptide, recombinant protein | Insulin | Sigma | Cat#: I6634 | |
| Peptide, recombinant protein | Recombinant Human/Murine/Rat BDNF | PeproTech | Cat#: 450–02 | |
| Peptide, recombinant protein | Recombinant Human NT-3 | PeproTech | Cat#: 450–03 | |
| Peptide, recombinant protein | Recombinant Murine EGF | PeproTech | Cat#: 315–09 | |
| Peptide, recombinant protein | Recombinant Human FGF-basic | PeproTech | Cat#: 100-18B | |
| Commercial assay or kit | Pierce BCA Protein Assay Kit | Thermo Fisher Scientific | Cat#: 23227 | |
| Commercial assay or kit | GeneSilencer Transfection Reagent | Genlantis | Cat#: T500750 | |
| Commercial assay or kit | Magna MeRIP m$^6$A Kit | Millipore | Cat#: 17–10499 | |
| Commercial assay or kit | EZ-Magna RIP RNA-Binding Protein Immunoprecipitation Kit | Millipore | Cat#: 17–701 | |
| Chemical compound, drug | cpt-cAMP, 8-(4-Chlorophenylthio) Adenosine 3':5'-CY | Sigma | Cat#: C3912 | |
| Chemical compound, drug | *N*-acetyl-L-cysteine (NAC) | Sigma | Cat#: A8199 | |
| Chemical compound, drug | Forskolin | Sigma | Cat#: F6886 | |
| Chemical compound, drug | Puromycin | Thermo Fisher Scientific | Cat#: A11138-03 | |
| Chemical compound, drug | Puromycin | Sigma | Cat#: P8833 | |
| Chemical compound, drug | Paraformaldehyde | Vetec | Cat#: V900894-100G | |
| Chemical compound, drug | Triton X-100 | Sigma | Cat#: V900502 | |
| Software, algorithm | GraphPad Prism 7.0 | GraphPad | https://www.graphpad.com, RRID: SCR_002798 | |

*Continued*

| Reagent type (species) or resource | Designation | Source or reference | Identifiers | Additional information |
|---|---|---|---|---|
| Software, algorithm | STAR v2.5 | *Dobin et al., 2013* | https://github.com/alexdobin/STAR/ RRID:SCR_004463 | |
| Software, algorithm | HTSeq | *Anders et al., 2015* | https://pypi.org/project/HTSeq/ | |
| Software, algorithm | ImageJ (Fiji) | *Schindelin et al., 2012* | http://fiji.sc, RRID:SCR_002285 | |
| Software, algorithm | Matlab | Matlab | https://ww2.mathworks.cn | |
| Other | TRIzol Reagent | Life | Cat#: 15596018 | |
| Other | PrimeScript RT Master Mix | Takara | Cat#: RR036B | |
| Other | 2× ChamQ Universal SYBR qPCR Master Mix | Vazyme | Cat#: Q711-02 | |
| Other | DMEM, high glucose | Gibco | Cat#: 11965–092 | |
| Other | Dulbecco's Modified Eagle's Medium, 10×, low glucose | Sigma | Cat#: D2429 | |
| Other | DMEM, high glucose | Hyclone | Cat#: SH30022.01 | |
| Other | Fetal Bovine Serum (FBS) | Gibco | Cat#: 10270–106 | |
| Other | Dulbecco's Phosphate-Buffered Saline, 1× without calcium and magnesium (DPBS) | Corning | Cat#: 21–031-CVR | |
| Other | Poly-D-lysine, Cultrex | Trevigen | Cat#: 3439-100-01 | |
| Other | Laminin (mouse), Culrex | Trevigen | Cat#: 3400-010-01 | |
| Other | DMEM/F-12, GlutaMAX | Gibco | Cat#: 10565–018 | |
| Other | Neurobasal Medium, minus phenol red | Gibco | Cat#: 12348–017 | |
| Other | Penicillin-Streptomycin | Life | Cat#: 15140–122 | |
| Other | B27 serum-free supplement, 50× | Life | Cat#: 17504044 | |
| Other | N-2 Supplement, 100× | Gibco | Cat#: 17502–048 | |
| Other | OCT Compound and Cryomolds, Tissue-Tek | SAKURA | Cat#: 4583 | |
| Other | ChemiBLOCKER | Millipore | Cat#: 2170 | |
| Other | CTB (Cholera Toxin Subunit B) conjugated by Alexa Fluor 555 | Invitrogen | Cat#: C34776 | |
| Other | VECTASHIELD Antifade Mounting Medium with DAPI | Vector Laboratory | Cat#: H-1200 | |
| Other | Mounting Medium, antifading (with DAPI) | Solarbio | Cat#: S2110 | |
| Other | Normal Goat Serum | Novus | Cat#: NBP2-23475 | |

## Animals and generation of the *Ythdf2* cKO mice

*Ythdf2^{fl/fl}* mice were reported previously (*Yu et al., 2021b*). *Six3-cre* (*Furuta et al., 2000*), *Thy1-GFP* (*Feng et al., 2000*), and *Rosa26-eYFP* (*Srinivas et al., 2001*) mice were from Jackson Laboratory. For timed pregnancy, embryos were identified as E0.5 when a copulatory plug was observed. Genotyping primers are as following: the first *Ythdf2-loxP* site, 5'-GCTTGTAGTTATGTTGTGTACCAC-3' and 5'-GCAGCTCTGACTATTCTAAAACCTCC-3'; the second *Ythdf2-loxP* site, 5'-CTCATAACATCCATAGCCACAGG-3', and 5'-CCAAGAGATAGCTTTCCTAATG-3'.

*Six3-cre* site, 5'-CCTTCCTCCCTCTCTATGTG-3' and 5'-GAACGAACCTGGTCGAAATC-3'.

*Rosa26-eYFP* wild type site, 5'-CTGGCTTCTGAGGACCG-3' and 5'-CAGGACAACGCCCACACA-3'; the mutant site, 5'-AGGGCGAGGAGCTGTTCA-3' and 5'-TGAAGTCGATGCCCTTCAG-3'. All experiments using mice were carried out following the animal protocols approved by the Laboratory Animal Welfare and Ethics Committee of Southern University of Science and Technology.

## Retinal neuronal culture

Retinal neurons were dissociated from E14.5 to 15.5 mouse embryos by papain in DPBS (1× Dulbecco's phosphate-buffered saline [PBS], Corning, NY) following the previously described methods (*Kechad et al., 2012*), and neuronal suspension was plated on acid-washed glass coverslips precoated with poly-D-lysine (Trevigen, 100 µg/ml) for 1 hr and laminin (Trevigen, 5 µg/ml) overnight at 37°C. Culture medium was made up of half DMEM/F12 medium (Gibco) and half neurobasal medium (Gibco), supplemented with B27 supplement (Life, 0.5×), penicillin-streptomycin (Life, 1×), N-2 supplement (Gibco, 0.5×), *N*-acetyl-L-cysteine (Sigma, NAC 0.6 mg/ml), cpt-cAMP (Sigma, 100 µM), forskolin (Sigma, 10 µM), and insulin (Sigma, 25 µg/ml). EGF (PeproTech, 50 ng/ml), BDNF (PeproTech, 50 ng/ml), NT-3 (PeproTech, 25 ng/ml), and FGF-basic (PeproTech, 10 ng/ml) were freshly added before using.

## KD using lentiviral shRNA, siRNA or AAV shRNA, and overexpression using AAV system

Lentiviral KD plasmids encoding shRNA (shCtrl: 5'-GCATCAAGGTGAACTTCAAGA-3'; sh*Ythdf2*: 5'-GGACGTTCCCAATAGCCAACT-3'; sh*Ythdf1*: 5'- GGACATTGGTACTTGGGATAA-3'; sh*Ythdf3*: 5'-GGATTTGGCAATGATACTTTG-3'; sh*Mettl14#6*: 5'-GCTGGACCTGGGATGATATTA-3'; sh*Mettl14#7*: 5'-CCCAGCTTGTACTTTGCTTTA-3') were generated from pLKO.1-TRC and lentivirus preparation process was described previously (*Yu et al., 2018*). All siRNAs were chosen from previous studies and the target sequences of siRNA are as following: siCtrl (RNAi negative control): 5'- UUCUCCGAACGU GUCACGUTT-3' (*Yu et al., 2018*); si*Kalrn7*: 5'- AGUACAAUCCUGGCCAUGUTT-3' (*Xie et al., 2007*); si*Kalrn9*: 5'-ACUGGACUGGACUUCUAUUTT-3' (*Yan et al., 2015*); si*Kalrn12*: 5'-CGAUGAGCUGAU GGAAGAATT-3' (*Yan et al., 2015*); si*Strn*: 5'-GGUGAAGAUCGAGAUACAATT-3' (*Breitman et al., 2008*); si*Ubr4*: 5'-AAUGAUGAGCAGUCAUCUAUTT-3' (*Shim et al., 2008*). AAV KD plasmids encoding shRNA (shCtrl: 5'-TTCTCCGAACGTGTCACGTAA-3'; sh*Kalrn12*: 5'-TGATGAGCTGATGGAAGAA-3'; sh*Ubr4*: 5'-AATGATGAGCAGTCATCTC-3') were generated using pHBAAV-U6-MCS-CMV-EGFP and packaged in serotype-9 by Hanbio (1.5 × $10^{12}$ genomic copies per ml). AAV overexpression plasmids of Hspa12a (NM_175199.3; PCR primer for mouse *Hspa12a*: 5'-ATGGCGGACAAGGAAGCTGG -3' and 5'-GTAATTTAAGAAGTCGATCCCC-3') and Islr2 (NM_001161541.1; PCR Primer for mouse *Islr2*: 5'-ATGGGGCCCTTTGGAGC-3' and 5'-GCCCGCTGTCTGCCTGTAG-3') were generated from pHBAAV-CMV-MCS-3flag-T2A-ZsGreen and packaged serotype-9 by Hanbio (1.2 × $10^{12}$ genomic copies per ml).

GeneSilencer Transfection Reagent (Genlantis) was used in siRNA transfection following the manufacturer's protocols. Culture medium was changed after 1 day of lentiviral shRNA infection or siRNA transfection. For lentiviral shRNA assay, puromycin (Thermo or Sigma, 1 µg/ml) was added after 2 days of infection. Immunofluorescence, RNA, or protein preparation was performed after shRNA or siRNA worked for 3 days. For AAV intravitreal injection, P0-P1 mouse pups were anesthetized in ice and then eyes were pierced at the edge of corneal by 30G × 1/2 needle (BD, 305106) under stereomicroscope. Then 1 µl AAV was intravitreally injected with 10 µl Syringe (Hamilton, 80330) following the pinhole. P15 or adult mice were anesthetized with 2.5% Avertin and then eyes were pierced at the side of corneal and the outer segment of sclera by 30G × 1/2 needle successively. Two µl AAV was intravitreally injected with 10 µl Syringe following the pinhole on the sclera. All subsequent experiments such as AOH operation and immunostaining were carried out after at least 3 weeks (10 days for ZsGreen/CART labeling of ooDSGCs in *Ythdf2* cKO and control mice in *Figure 3A*).

## RT-qPCR

Total RNA was extracted from cells or tissues with TRIzol Reagent (Life) and then used for reverse transcription by PrimeScript RT Master Mix (TaKaRa). Synthesized cDNA was used for qPCR by 2× ChamQ Universal SYBR qPCR Master Mix (Vazyme) on StepOnePlus Real-Time PCR System (ABI) or BioRad CFX96 Touch Real-Time PCR system. Primers used for qPCR are as following: mouse *Gapdh*: 5'-TTGTCAGCAATGCATCCTGCACCACC-3' and 5'-CTGAGTGGCAGTGATGGCATGGAC-3' (*Mains et al., 2011*); mouse *Kalrn7*: 5'- GATACCATATCCATTGCCTCCAGGACC-3' and 5'-CCAGGCTGCGCG CTAAACGTAAG-3' (*Mains et al., 2011*); mouse *Kalrn9*: 5'- GCCCCTCGCCAAAGCCACAGC-3' and 5'-CCAGTGAGTCCCGTGGTGGGC-3' (*Mains et al., 2011*); mouse *Kalrn12*: 5'- CAGCAGCCACGT GCCTGCAGC-3' and 5'-TCTTGACATTGGGAATGGGCCGCAC-3' (*Mains et al., 2011*); mouse

*Strn*: 5′-TGAAGCCTGGAATGTGGACC-3′ and 5′-CTATTGGGCCTCTTCACCCC-3′; mouse *Ubr4*: 5′- TGAGTGAGGACAAGGGCAAC-3′ and 5′-GGGTTGGATCGAACGAAGGT-3′; mouse *Ythdf2*: 5′-GAGCAGAGACCAAAAGGTCAAG-3′and 5′-CTGTGGGCTCAAGTAAGGTTC-3′; 18 s: 5′-GCTTAATT TGACTCAACACGGGA-3′ and 5′-AGCTATCAATCTGTCAATCCTGTC-3′ (*Wang et al., 2018*); mouse *Hspa12a*: 5′-GGGTTTGCACAGGCTAAGGA-3′ and 5′-TCTGATGGACGGTCAGGTCT-3′; mouse *Islr2*: 5′-GAAGCTCCCTTAGACTGTCACC-3′ and 5′-CCCCATCGTGACTCCTGCTG-3′.

## Immunofluorescence and immunostaining

For tissue sections, mouse embryonic eyes were fixed with 4% PFA (Sigma) in 0.1 M phosphate buffer (PB) for 30–45 min at room temperature (RT); eyes of mouse pups (<P10) were pre-fixed briefly and then eyecups were dissected and fixed for 45 min-1 hr at RT; for P20-30 or adult mice, eyecups were dissected after myocardial perfusion with 0.9% NaCl, followed by fixation for 1 hr. After PBS (3 × 5 min) washing, tissues were dehydrated with 30% sucrose in 0.1 M PB overnight at 4°C, then embedded with OCT (SAKURA) and cryosectioned at 12 µm (20 µm for Thy1-GFP section analysis) with Leica CM1950 Cryostat. Tissue sections were permeabilized and blocked with 10% ChemiB-LOCKER (Millipore) and 0.5% Triton X-100 (Sigma) in PBS (PBST) for 1 hr at RT and incubated in PBST overnight at 4°C with following primary antibodies: chicken anti-GFP (1:1000, Abcam ab13970), chicken anti-MAP2 (1:10,000, Abcam ab5392), goat anti-VAChT (1:1000, Millipore ABN100), guinea pig anti-RBPMS (1:1000, PhosphoSolutions 1832-RBPMS), mouse anti-AP2α (1:1000, DSHB 3B5), mouse anti-Bassoon (1:2500, Enzo Life Sciences ADI-VAM-PS003), mouse anti-Brn3a (1:300, Millipore MAB1585), mouse anti-Calbindin-D-28K (1:200, Sigma C9848), mouse anti-PKCα (1:500, Santa Cruz sc-8393), rabbit anti-Strn (Striatin) (1:500, BD Biosciences 610838), rabbit anti-CART (1:2000, Phoenix Pharmaceuticals H-003–62), rabbit anti-m$^6$A (1:200, Synaptic Systems 202003), rabbit anti-melanopsin (1:1000, Thermo PA1-780), rabbit anti-PKCα (1:1000, Cell Signaling CST-2056), rabbit anti-PSD95 (1:1000, Abcam ab18258), rabbit anti-Recoverin (1:1000, Millipore AB5585), rabbit anti-YTHDF2 (1:1000, Proteintech 24744–1-AP), rabbit anti-YTHDF1 (1:1000, Proteintech 17479–1-AP), rabbit anti-YTHDF3 (1:1000, Abcam ab103328), rabbit anti-Ubr4 (1:300, Abcam ab86738), sheep anti-Chx10 (1:1000, Exalpha X1179P), chicken anti-GFAP (1:500, Millipore AB5541), goat anti-Lhx2 (1:200, Santa Cruz Biotechnology sc-19344), rabbit anti-Lhx2 (1:500, Abcam ab184337). After three times of PBS washing, sections were incubated in PBST for 1 hr at RT with secondary antibodies: Alexa 488 donkey anti-chicken (1:500, Jackson 703-545-155), Alexa 488 donkey anti-guinea pig (1:500, Jackson 706-545-148), Alexa 488 donkey anti-mouse (1:500, Thermo A21202), Alexa 488 donkey anti-rabbit (1:500, Thermo A21206), Alexa 555 donkey anti-goat (1:1000, Thermo A21432), Alexa 555 donkey anti-mouse (1:1000, Thermo A31570), Alexa 555 donkey anti-rabbit (1:1000, Thermo A31572), Alexa 555 donkey anti-sheep (1:1000, Thermo A21436), Alexa 555 goat anti-chicken (1:1000, Thermo A21437), or Alexa 647 donkey anti-mouse (1:200, Thermo A31571) and then mounted with the VECTASHIELD Antifade Mounting Medium with DAPI (Vector Laboratory).

For cultured neurons, after twice of PBS washing, cells were fixed for 15 min with 4% PFA in 0.1 M PB at RT, then washed with PBS three times and blocked in PBST for 20 min at RT. Antibody incubation conditions are the same as tissue sections.

For wholemount immunostaining of retina, eyes were dissected after myocardial perfusion with 0.9% NaCl. Then retinas were separated from sclera and fixed with 4% PFA in 0.1 M PB for 1 hr at RT. Then retinas were blocked with 5% normal goat serum (Novus), 0.4% Triton X-100 in PBS overnight at 4°C. Primary antibodies such as chicken anti-GFP (1:1000, Abcam ab13970), mouse anti-Brn3a (1:300, Millipore MAB1585), mouse anti-SMI-32 (1:200, BioLegend 801701), or rabbit anti-Melanopsin (1:1000, Thermo PA1-780), rabbit anti-CART (1:2000, Phoenix Pharmaceuticals H-003–62) were diluted in 5% normal goat serum, 0.4% Triton X-100 in PBS and incubated overnight at 4°C. Then retinas were incubated with Alexa 488 donkey anti-chicken (1:500, Jackson 703-545-155), Alexa 488 donkey anti-mouse (1:500, Thermo A21202), Alexa 555 donkey anti-mouse (1:1000, Thermo A-31570) and Alexa 555 donkey anti-rabbit (1:1000, Thermo A31572) secondary antibodies in 5% normal goat serum (Novus), 0.4% Triton X-100 in PBS and finally mounted with the VECTASHIELD Antifade Mounting Medium with DAPI.

All images were captured on Nikon A1R confocal microscope or Zeiss LSM 800 confocal microscope with identical settings for each group in the same experiment. A region of interest, length or thickness in immunofluorescence experiments were obtained with ImageJ. The number of neurons

in specific area was counted blindly and manually. To quantify RGC dendrite lamination in IPL with Thy1-GFP, z-stack and maximum projection were performed during the analysis. GFP intensity values across IPL depth were measured by ImageJ/Analyze/Plot Profile function (*Liu et al., 2018*). To quantify the numbers of Bassoon[+]/PSD-95[+] excitatory synapses in IPL, the colocalization puncta was measured by ImageJ/Analyze/Puncta Analyzer as described previously (*Ippolito and Eroglu, 2010*).

## Sholl analysis

For confocal images of cultured RGCs, MAP2 signals in original format were analyzed with simple neurite tracer and then quantified with Sholl analysis (5 μm per distance from soma center) which was a widely used method in neurobiology to quantify the complexity of dendritic arbors using ImageJ (*Schindelin et al., 2012*; *Binley et al., 2014*). Retina wholemount data were captured in z-stack mode (0.5–1 μm per slide) with confocal microscopes. ZsGreen, eGFP, and SMI-32 signals were directly analyzed with simple neurite tracer and then z projection of all tracers was quantified with Sholl analysis (10 μm per distance from soma center), while melanopsin signals were maximum-projected before tracing.

## OMR assay

*Ythdf2* cKO and control mice aged about 6 weeks were dark-adapted overnight before experiment and used in the OMR assay following the previously reported protocols (*Douglas et al., 2005*; *Sergeeva et al., 2018*). Using the Matlab program, 0.2 c/deg (15 s per direction of rotation) was first used for mice to adapt this experiment, and 0.3, 0.35, 0.4, 0.43, 0.45, 0.47, 0.5 and 0.55 c/deg (30 s per direction of rotation) were used in the following recordings. Mouse behaviors were analyzed in real time during the experiment and re-checked with video recordings. Finally, data for each mouse were determined by the minimal spatial frequency between left and right OMR.

## CTB labeling of optic nerve

To label RGC axon terminals in mouse brain, RGC axons were anterogradely labeled by CTB conjugated with Alexa Fluor 555 (Invitrogen, C34776) through intravitreal injection 48 hr before sacrifice. After PFA perfusion, the brains were fixed with 4% PFA in 0.1 M PB overnight, dehydrated with 15% sucrose and 30% sucrose in 0.1 M PB overnight at 4°C sequentially, embedded with OCT for coronal section, and cryosectioned at 12 μm with Leica CM1950 Cryostat. After PBS washing, the sections were mounted with VECTASHIELD Antifade Mounting Medium with DAPI (Vector Laboratory). The images were captured on Tissue Genostics with identical settings for each group in the same experiment with the TissueFAXS 7.0 software.

## RIP and sequencing

For RIP experiment, we used the EZ-Magna RIP RNA-Binding Protein Immunoprecipitation Kit (Millipore) following the manual with minor modifications. Briefly, $1 \times 10^7$ retinal neurons were subjected to each 100 μl lysis buffer. The amount of YTHDF2 antibody (Proteintech, 24744–1-AP) and control IgG used for immunoprecipitation is 5 μg, respectively. RIP experimental steps, RNA sample preparation and sequencing, and sequence data analysis followed the procedures reported previously (*Yu et al., 2021a*).

## MS analysis

E15.5 retinal neurons were cultured and infected with lentiviral sh*Ythdf2* or shCtrl. Sample collection and lysis, protein and peptide preparation were performed following procedures reported previously (*Yu et al., 2021b*). Proteins with fold changes greater than 1.3 and p values less than 0.05 were considered to be regulated by YTHDF2 KD with statistical significance.

## Anti-m⁶A immunoprecipitation

Total retinal RNA was extracted from P0 WT mouse pups. Immunoprecipitation of m⁶A-modified transcripts was carried out with Magna MeRIP m⁶A Kit (Merck-Millipore, 17–10499) following the manual. m⁶A antibody (Synaptic Systems, 202003) and corresponding control IgG were used in this experiment. The RNA samples pulled down from the experiment were used for RT-qPCR.

## AOH model

Mice were anesthetized with 5% chloral hydrate in normal saline (10 μl/g) based on body weight and the Compound Tropicamide Eye Drops were used to scatter pupil. The anterior chamber was penetrated using the 32G × 1/2" needles (TSK) and filled with the BBS Sterile Irrigating Solution (Alcon) which was hung at a high position to provide proper pressure. Intraocular pressure was measured with the Tonolab tonometer (icare) for every 10 min and maintained at 85–90 mmHg for 1 hr. Levofloxacin hydrochloride was used after the operation and mice were revived in a 37°C environment. Retinas were analyzed for gene expression of YTHDF2 1 day after AOH, gene expression of *Hspa12a* and *Islr2* 3 days after AOH, dendritic complexity and RGC number 3–7 days after AOH.

## Statistical analysis

All experiments were conducted at a minimum of three independent biological replicates (two biological replicates for the RIP assay) or three mice/pups for each genotype/condition in the lab. Data are mean ± SEM. Statistical analysis was preformed using GraphPad Prism 7.0. When comparing the means of two groups, an unpaired or paired $t$ test was performed on the basis of experimental design. The settings for all box and whisker plots are: 25th-75th percentiles (boxes), minimum and maximum (whiskers), and medians (horizontal lines). A p value less than 0.05 was considered as statistically significant: $*p < 0.05$, $**p < 0.01$, $***p < 0.001$, $****p < 0.0001$.

## Acknowledgements

We thank Ke Wang and Kwok-Fai So (Jinan University) for help on the OMR assay. We thank Mengqing Xiang and Suo Qiu (Zhongshan Ophthalmic Center, Sun Yat-sen University) for help on AAV experiments. We thank other members of Ji laboratory for technical support, helpful discussions, and comments on the manuscript. This work was supported by National Natural Science Foundation of China (31871038 and 32170955 to S-JJ; 31922027 and 32170958 to BP), Shenzhen-Hong Kong Institute of Brain Science-Shenzhen Fundamental Research Institutions (2021SHIBS0002, 2019SHIBS0002), High-Level University Construction Fund for Department of Biology (internal grant no. G02226301), Science and Technology Innovation Commission of Shenzhen Municipal Government (ZDSYS20200811144002008), Program of Shanghai Subject Chief Scientist (21XD1420400), and the Innovative Research Team of High-Level Local University in Shanghai (BP).

## Additional information

### Funding

| Funder | Grant reference number | Author |
|---|---|---|
| National Natural Science Foundation of China | 31871038 | Sheng-Jian Ji |
| National Natural Science Foundation of China | 31922027 | Bo Peng |
| Shenzhen-Hong Kong Institute of Brain Science-Shenzhen Fundamental Research Institutions | 2021SHIBS0002 | Sheng-Jian Ji |
| High-Level University Construction Fund for Department of Biology | G02226301 | Sheng-Jian Ji |
| Science and Technology Innovation Commission of Shenzhen Municipality | ZDSYS20200811144002008 | Sheng-Jian Ji |
| Program of Shanghai Subject Chief Scientist | 21XD1420400 | Bo Peng |

| Funder | Grant reference number | Author |
|---|---|---|
| Innovative Research Team of High-Level Local University in Shanghai | | Bo Peng |
| Shenzhen-Hong Kong Institute of Brain Science-Shenzhen Fundamental Research Institutions | 2019SHIBS0002 | Sheng-Jian Ji |
| National Natural Science Foundation of China | 32170955 | Sheng-Jian Ji |
| National Natural Science Foundation of China | 32170958 | Bo Peng |

The funders had no role in study design, data collection and interpretation, or the decision to submit the work for publication.

## Author contributions

Fugui Niu, Conceptualization, Data curation, Formal analysis, Investigation, Methodology, Validation, Visualization, Writing – original draft, Writing – review and editing; Peng Han, Data curation, Formal analysis, Investigation, Methodology, Validation, Visualization; Jian Zhang, Yuanchu She, Lixin Yang, Jun Yu, Formal analysis, Investigation, Methodology; Mengru Zhuang, Kezhen Tang, Investigation, Methodology; Yuwei Shi, Baisheng Yang, Investigation; Chunqiao Liu, Investigation, Resources; Bo Peng, Conceptualization, Methodology, Resources, Software, Supervision, Writing – review and editing; Sheng-Jian Ji, Conceptualization, Funding acquisition, Project administration, Resources, Supervision, Writing – original draft, Writing – review and editing

## Author ORCIDs

Bo Peng http://orcid.org/0000-0003-4183-5939
Sheng-Jian Ji http://orcid.org/0000-0003-3380-258X

## Ethics

All experiments using mice were carried out following the animal protocols approved by the Laboratory Animal Welfare and Ethics Committee of Southern University of Science and Technology (approval numbers: SUSTC-JY2017004, SUSTC-JY2019081).

## Decision letter and Author response

Decision letter https://doi.org/10.7554/eLife.75827.sa1
Author response https://doi.org/10.7554/eLife.75827.sa2

# Additional files

## Supplementary files

- Supplementary file 1. List of YTHDF2 target mRNAs by anti-YTHDF2 RIP-Seq.
- Supplementary file 2. Proteome of YTHDF2 knockdown vs. control.
- Supplementary file 3. Overlapping mRNA of Y2-RIP vs. Y2-KD-MS.
- Transparent reporting form

## Data availability

The RIP-seq data have been deposited to the Gene Expression Omnibus (GEO) with accession number GSE145390. The mass spectrometry proteomics data have been deposited to the ProteomeXchange Consortium via the PRIDE partner repository with the dataset identifier PXD017775.

The following datasets were generated:

| Author(s) | Year | Dataset title | Dataset URL | Database and Identifier |
|---|---|---|---|---|
| Niu F, Yang L, Ji S | 2022 | Anti YTHDF2 RIP-seq to identify YTHDF2 target mRNAs in P0 mouse retinas | https://www.ncbi.nlm.nih.gov/geo/query/acc.cgi?acc=GSE145390 | NCBI Gene Expression Omnibus, GSE145390 |
| Niu F, Ji S | 2022 | Proteome analysis using mass spectrometry (MS) in acute shYthdf2-mediated knockdown of cultured RGCs | https://www.ebi.ac.uk/pride/archive/projects/PXD017775 | PRIDE, PXD017775 |

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
