## [Editor Report]

In this study, you propose a role for the m^6^A reader YTHDF2 in regulating the dendritic arbor size of retinal ganglion cells (RGCs). You show that retina-specific loss of *Ythdf2* leads to the expansion of RGC dendritic arbors in the horizontal plane and a widening of the inner plexiform layer (IPL), with *Ythdf2* conditional knockouts showing a modest increase in visual acuity in an optomotor assay. You point to a number of factors downstream of YTHDF2 not previously known to be involved in retinal dendritic development, and propose a role for YTHDF2 in a glaucoma model, in which loss of YTHDF2 is shown to prevent RGC loss. This study presents a careful phenotypic analysis of manipulation of YTHDF2 and provides a foundation for studies on how YTHDF2-mediated mechanisms are integrated into programs of dendritic development and RGC survival.

---

## [Decision Letter]

**Decision letter after peer review:**

[Editors’ note: the authors submitted for reconsideration following the decision after peer review. What follows is the decision letter after the first round of review.]

Thank you for submitting your work entitled "The m6A reader YTHDF2 is a negative regulator for dendrite development and maintenance of retinal ganglion cells" for consideration by *eLife*. Your article has been reviewed by 3 peer reviewers, and the evaluation has been overseen by a Reviewing Editor and a Senior Editor. The reviewers have opted to remain anonymous.

Our decision has been reached after consultation between the reviewers. Based on these discussions and the individual reviews below, we regret to inform you that your work will not be considered further for publication in *eLife*.

The reviewers have provided thorough assessment of your study proposing a role for the m6A reader YTHDF2 in regulating retinal ganglion cell (RGCs) dendritic arbor size, and maintenance of RGCs, with implications for retinal IPL morphology and for glaucoma. Unfortunately, as you can read in the reviews, the reviewers had a number of concerns that would each require additional work in a longer time frame required by *eLife* for revision (and recognizing the additional downtime due to the COVID crisis). The concerns include:

– Better phenotyping of the Ythdf2 cKO mouse model, including cross sections of retina in addition to horizontal views to more precisely measure layer dimensions, synapse number and dendritic extension; whether cells other than RGCs are affected; co-localization of YTHDF2 and RGC markers. Moreover, the connection between the dendrite development phenotype and the neuroprotective role of YTHDF2 is not clear from your analyses.

– in vivo evidence for the putative targets of YTHDF2, rather than simply in vitro, a better understanding of how the products produced by these mRNAs – kalirin, Strn and Ubr4 – contribute to the increase in dendrite branching in the Ythdf2 knockout; the mechanism of YTHDF2 upregulation induced by AOH and loss of Brn3a+ RGCs thereof.

– The use of the optomotor response is not an apt measure function, especially in the absence of connecting the response to the classes of direction-selective RGCs which one would expect might interface with OMR responses.

– Ythdf2 effects in the acute ocular hypertension (AOH) model: Is this related to the downstream mRNAs, or some other more general effect of loss of this m6 reader? And how is this related to RGC survival at a mechanistic level (and thus the relation to glaucoma)?

We hope that you can use the reviewer's comments to amend your manuscript for submission elsewhere.

*Reviewer #1:*

The authors report a functional-relevant role of YTHDF2 in mouse retina ganglion cells (RGCs) to negatively regulate dendrite development and maintenance of RGCs. Mechanistically, Kalrn, Strn, and Ubr4 mRNAs are selectively recognized and decay through the YTHDF2-mediated pathway, the subsequent downregulation of proteins encoded by those mRNAs negatively regulate dendrite development and maintenance of RGCs.

This work highlights regulatory roles of N6-methyladenosine (m6A) and YTHDF2 protein in retina and suggested YTHDF2 as one possible target in AOH-induced RGC loss. A few concerns will need to be addressed.

Concerns:

1. The authors showed YTHDF2 mRNA is upregulated by AOH and argued a loss of YTHDF2 in retina results in less severe AOH-induced loss of Brn3a+ RGCs. The mechanism of YTHDF2 upregulation induced by AOH and loss of Brn3a+ RGCs mediated by YTHDF2 upregulation are unclear. The first question might be harder to address but the second one is within the scope of this work.

2. In figure 7B, YTHDF2 expression in Brn3a+ RGCs was quantified. The figure showed that Brn3a was not detected in some cells. In figure 7C, mRNA level was quantified with an extraction method that did not separate Brn3a+ and Brn3a- cells.

3. The authors showed AOH-induced loss of Brn3a+ RGCs in mouse models. Supportive evidence at the phenotype level such as less severe loss of visual acuity in YTHDF2 cKO mice are required.

4. The authors showed knockdown of Kalrn, Strn, and Ubr4 are sufficient to impair dendrite growth. Rescue experiments will further back up their claim.

5. The authors suggested that m6A-YTHDF2 axis regulates the stability of Kalrn, Strn, and Ubr4 mRNAs. The authors will need to show these RNAs contain m6A by m6A-IP-qPCR. The effect on Kalrn, Strn, and Ubr4 mRNAs upon m6A writer complex knockdown can also be shown.

*Reviewer #2:*

This manuscript investigates the role of the m6A reader YTHDF2 during retinal ganglion cell (RGC) dendrite development and branching. The data presented in this study shows that deletion of YTHDF2 in the retina led to increased RGC dendrite branching and improved visual acuity. Based on high throughput analysis, YTHDF2 is suggested to regulate dendrite branching by reducing the stability of kalirin, Strn and Ubr4 mRNAs. Lastly, loss of YTHDF2 is shown to prevent RGC loss in a glaucoma model. The experimental designs were presented clearly, and the results were interpreted with the appropriate statistical tests. However, the phenotype of the Ythdf2 cKO mouse model is not well characterized and there lacks in vivo evidence for the putative targets of YTHDF2. It is also difficult to explain the connection, if any, between the dendritic role of YTHDF2 and its neuroprotective function.

1. The authors used RGC cultures to show that knockdown/knockout of YTHDF2 resulted in increased the dendrite branching based on the number of interactions in Sholl analysis. However, to determine the full extent of dendritic defect, it is important to perform detailed analysis of the dendrite structure, including the total dendritic length, number of total segments, maximum branch length, branch order and number of branches.

2. It is suggested in the manuscript that loss of YTHDF2 in the retina only affects dendrite development of RGCs, but not other cell types. This is based on the simple counting of the number of RPCs at E15 and bipolar and amacrine cells at P15, glossing over the vast complexity of the visual circuitry in the retina. At the very least, the authors should confirm that differentiation of all major retinal neurons and proper formation of the ribbon synapses were unaffected in the cKO mouse model of YTHDF2, which would other confound the interpretation of the RGC and visual acuity phenotype.

3. The in vivo characterisation of RGC dendrite morphology should be more thorough as previously mentioned, including the arbor shape, size and tiling pattern. The image quality of Melanopsin and SMI-32 staining is low. There are too many cells labelled, making it difficult to examine individual cells/dendrites. Higher resolution images from more sparsely labelled retina would be needed.

4. The authors provide evidence of increased synaptic density in the IPL using PSD-95 staining in the cKO mice compared to the control litter mates. It is important to investigate if there are any changes to the synaptic density in the OPL. In addition, it is also important to study the number of successful synapse formation by co-staining the tissues with presynaptic markers such as bassoon with PSD-95 and count for co-localisation.

5. The improved visual acuity in Six3Cre-mediated Ythdf2 cKO is solely attributed to the RGC dendrite defects. However, Six3Cre is widely expressed in the ventral telencephalon and ventral anterior hypothalamus, including the SCN and SPZ (PMID: 24767996). It is necessary to examine the retinocollicular projections of the RGC axons.

6. The Optomotor response (OMR)-based assay is not the most appropriate test to test the response of ganglion cells since the improved OMR could be attributed to other cell types as previously mentioned. A more targeted approach such as pattern ERG could provide more evidence to their claims.

7. The quality of the RIP-seq analysis is not clear. Is there an isotype antibody control? Are there biological replicates? What is the statistical significance of the top hits? There are similar concerns regarding the MS analysis and whether there is significant overlap between the target lists generated from RIP-Seq and MS studies.

8. The authors provide evidence in the RGC culture that stabilisation of kalirin, Strn and Ubr4 mRNAs could contribute to the increase in dendrite branching in Ythdf2 knockout. The regulation of these genes by Ythdf2 need to be validated in Ythdf2 cKO retina at the level of mRNA by qRT-PCR and at the level of protein by western blot. Functionally, the authors need to investigate if knockdown or knockout of these genes in retina indeed affect dendrite branching or even rescue Ythdf2 cKO phenotype.

9. Finally, there is not clear connection between the dendrite development phenotype and the neuroprotective role of YTHDF2. The authors need to clarify the rationale behind these experiments.

*Reviewer #3:*

In this study, the authors propose a role for the m6A reader YTHDF2 in regulating the dendritic arbor size of retinal ganglion cells (RGCs), and they extrapolate this to address issues in glaucoma and overall retina IPL morphology. They find that retina-specific loss of Ythdf2 leads to the expansion of RGC dendritic arbors in the horizontal plane and also a widening of the inner plexiform layer (IPL), with Ythdf2 cKOs showing a modest increase in visual acuity in an optomotor assay. They identify a number of factors downstream of YTHDF2 not previously known to be involved in retinal dendritic development, and they propose a role for Ythdf2 in a glaucoma model, providing a foundation for future studies into their functions in these contexts.

Although these data are sufficient to establish a role for Ythdf2 in RGC dendritic development and other more general aspects of retina morphology, many of the authors' specific claims are weakened by issues with the experiments, their analyses, and by weak effect sizes. Even if the points raised below were addressed, it remains unclear from this study how Ythdf2-mediated mechanisms are integrated into programs of dendritic development and RGC survival, diminishing the impact of this work. This study would be strengthened by, for example, identifying a developmental program that modifies the target mRNAs found in this study to promote recognition by YTHDF2, and also by providing better understanding of how the products produced by these mRNAs actually regulate dendrite morphology and synapse formation. Overall, while there are a number of promising findings in this study, the work required to address weaknesses falls well outside of the limits usually recommended by *eLife*, and it is suited for publication in a more specialized journal following attention to many of the points raised below.

1. The cell-type specific tracing in Figure 3 is difficult to assess due to the extremely high background in the melanopsin images and unclear signal in the SMI-32 images. This analysis would be improved by sparse viral labeling, with a fluorophore provided by the virus used for tracing and the melanopsin & SMI-32 antibodies used solely for establishing RGC subtype identity. Further, the authors emphasize that the defects they observe here are in the plane of retinal ganglion cell (RGC) dendritic arbors, but this is an over-interpretation of these data. One wonders why the authors did not examine in cross section the lamination within the IPL of the melanopsin-positive ipRGCs, since this would provide, at least for this RGC subclass, a direct assessment of this issue. Finally, the Thy-1-O line GFP distribution is quite general, and it really is not the correct line to use if one wants to investigate defects in lamination.

2. The authors repeatedly describe increased dendritic and synaptic density within the IPL in Ythdf2 cKOs, yet they only present absolute signal intensities to support these claims; without normalizing signal intensity to IPL width, it is impossible to determine whether changes in intensity are indeed due to increased density or from the IPL becoming enlarged with a similar density of dendrites and synapses. Further, they claim in the text that VGlut2 puncta numbers are changed in these mutants, however the analysis presented is only for general signal intensity. If claims are to be made regarding changes in synaptic specializations, higher resolution assessments of individual puncta using pre- and postsynaptic markers is required.

3. While Ythdf2 cKOs do show a statistically significant improvement in visual acuity (although for these male data it is unclear whether the control data distribution is sufficiently normal to use a t-test), the effect size is small enough to call into question how meaningful the improvement is. Further, what is the actual relationship between this modest improvement in optomotor response (OMR) and the morphological phenotypes observed here, in particular since they are not connected here to the classes of direction-selective RGCs one would expect might affect OMR responses.

4. Much clearer co-localization between YTHDF2 and RGC markers is needed, both in vivo and in vitro, to clearly demonstrate association with RGCs and not displaced amacrine cells or other retinal cell types.

5. The several mRNAs and their produces shown to be under YTHDF2 control do appear to play some role in regulating general RGC dendritic morphology, though of course from these data we do not know whether or not there is any RGC-subtype specificity. However, as presented, we really do not learn much at all about their mechanisms of action. At the very least, do they act in the same, or different, pathways….i.e. have any attempts been made to knock down more than one at a time and ask about independent pathway affiliations?

6. It is not clear how to interpret the Ythdf2 effects in the acute ocular hypertension (AOH) model. Is this related to the downstream mRNAs, or some other more general effect of loss of this m6 reader? And how is this related to RGC survival at a mechanistic level?

[Editors’ note: further revisions were suggested prior to acceptance, as described below.]

Thank you for resubmitting your work entitled "The m^6^A reader YTHDF2 is a negative regulator for dendrite development and maintenance of retinal ganglion cells" for further consideration by *eLife*. Your revised article has been evaluated by Catherine Dulac (Senior Editor) and Carol Mason as Reviewing Editor.

The manuscript has been improved but there are some remaining issues that need to be addressed, as outlined below and in the appended reviews:

Reviewer 2 (previous Reviewer 1) was satisfied with the revisions of this resubmission as was Reviewer 3 (previous Reviewer (3) although the latter reviewer adds that "…we still are far from a mechanistic understanding of how these gene products regulate RGC dendritic arborization, and whether or not there is any specificity of these effects with regard to RGC subtypes."

Reviewer 1 (previous Reviewer 2) continues to have concerns: He summarized in the Public Review (this will not be made public as it is a resubmission) that "the phenotype of the Ythdf2 cKO mouse model is not well characterized and there lacks in vivo evidence for the putative targets of YTHDF2".

Two items he previously requested were not addressed: (1) to perform either qPCR or western blots of Ythdf2 cKO retina to confirm the regulation of Kalirin, Strn and Ubr4 by Ythdf2, and (2) to show that overexpression of Hspa12a and Islr2 alleviates loss of RGCs in the AOH model (See point #8 of previous Reviewer 2’s review). We ask that you address these two items that were raised in the first review, to the best of your ability, given time constraints. (Thus, these are not experiments newly requested by the reviewer(s)).

Current Reviewer 1 has also raised additional points not in the first review, that could be answered mostly textually:

1. Your stated exclusivity of YTHDF2 expression in RGCs, given that there is apparent staining in all cell layers in Suppl Figure 1B. He cited very "old"scRNAseq data indicating that YTHDF2 is found in all retinal cell types.

2. To verify that YTHDF2 is neuronal: determining YTHDF2 expression in Muller glia and astrocytes.

3. Addressing the utility of the Six3-Cre, by determining whether there is mosaic expression of Cre in the peripheral retina.

*Reviewer #1:*

The experimental designs were presented clearly and the results were interpreted with the appropriate statistical tests. However, the phenotype of the Ythdf2 cKO mouse model is not well characterized and there lacks in vivo evidence for the putative targets of YTHDF2. It is also difficult to explain the connection, if any, between the dendritic role of YTHDF2 and its neuroprotective function. Most importantly, whether Ythdf2 acts cell autonomously in the RGCs remains to be resolved.

1. The premise of this study is that Ythdf2 is cell autonomously required in RGCs for dendritic development. In the rebuttal letter, the authors even went as far as claiming that "all RGCs express YTHDF2 and all YTHDF2-expressing cells in retina are RGCs". However, this statement is belied by the strong YTHDF2 staining outside the RGC layers in Figure 1—figure supplement 1B. The YTHDF2 expression in non-RGC cells can also be seen in Figure 1A, although for unknown reason the staining intensity was weaker. In fact, numerous RNAseq studies have now thoroughly catalogued gene expression patterns in the retina throughout development and YTHDF2 expression can be found among all retinal cell types even from the earliest scRNAseq data (Macosko et al., 2015, Cell 161, 1202). The ubiquitous expression of YTHDF2 in the retina poses serious concerns regarding the usefulness of Six3-Cre mediated Ythdf2 cKO model and the validity of presumed Ythdf2 downstream targets in RGCs.

2. The revised manuscript presents data to show that retinal progenitors, amacrine cells, bipolar cells, photoreceptors, or horizontal cells were not affected in Ythdf2 cKO retina. It is also important for the authors to examine Müller glia and astrocyte, especially the latter which is known to have a strong influence on RGC activity and health.

3. The potential targets of YTHDF2 were mostly characterized in the RGC culture, which is not a good model for studying dendritic development because RGCs will evitable die in vitro. As previously requested by this reviewer, the authors need to perform either qPCR or western blos of Ythdf2 cKO retina to confirm the regulation of Kalirin, Strn and Ubr4 by Ythdf2. They also need to show whether siRNA knockdown or CRISPR knockout of these genes rescue the Ythdf2 cKO phenotype in vivo.

4. The authors propose that the better maintenance of RGC dendrite arborization in Ythdf2 cKO retina make it resistant to RGC death in AOH model and this is mediated by Hspa12a and Islr2 genes. To support this claim, the authors need to show that overexpression of Hspa12a and Islr2 alleviates loss of RGCs in the AOH model.

*Reviewer #2:*

The authors have responded to my comments and suggestions with several new experiments and some additional explanations. Though the effects of YTHDF2 described here are pleiotropic across RGC cell types, these defects in dendritic arborization are better described in this revised manuscript. In addition, further attention to the mRNAs regulated by YTHDF2 is given, and though we still are far from a mechanistic understanding of how these gene products regulate RGC dendritic arborization, and whether or not there is any specificity of these effects with regard to RGC subtypes, I find this revision to be a substantial improvement over the initial submission.

*Reviewer #3:*

The authors have addressed my comments.

---

## [Author Response]

[Editors’ note: the authors resubmitted a revised version of the paper for consideration. What follows is the authors’ response to the first round of review.]

Reviewer #1:The authors report a functional-relevant role of YTHDF2 in mouse retina ganglion cells (RGCs) to negatively regulate dendrite development and maintenance of RGCs. Mechanistically, Kalrn, Strn, and Ubr4 mRNAs are selectively recognized and decay through the YTHDF2-mediated pathway, the subsequent downregulation of proteins encoded by those mRNAs negatively regulate dendrite development and maintenance of RGCs.This work highlights regulatory roles of N6-methyladenosine (m6A) and YTHDF2 protein in retina and suggested YTHDF2 as one possible target in AOH-induced RGC loss. A few concerns will need to be addressed.Concerns:1. The authors showed YTHDF2 mRNA is upregulated by AOH and argued a loss of YTHDF2 in retina results in less severe AOH-induced loss of Brn3a+ RGCs. The mechanism of YTHDF2 upregulation induced by AOH and loss of Brn3a+ RGCs mediated by YTHDF2 upregulation are unclear. The first question might be harder to address but the second one is within the scope of this work.

We agree with the reviewer that the mechanism of YTHDF2 upregulation induced by AOH is beyond the scope of this work, which may require further investigation as a future direction.

As for the mechanism of loss of Brn3a^+^ RGCs mediated by YTHDF2 upregulation in AOH-treated retina, we tried to answer this question by expanding the scope of our study in the revised manuscript. We wanted to know whether YTHDF2 target mRNAs mediate these effects in the AOH models. AOH has been shown to cause pathological changes in RGC dendrites before axon degeneration or soma loss is detected in different model animals (Morgan et al., 2006; Shou et al., 2003; Weber et al., 1998), suggesting that RGC soma loss might be following dendrite degeneration. So we tested whether YTHDF2 upregulation mediates RGC dendrite degeneration in AOH-treated retina. We found that two target mRNAs *Hspa12a* and *Islr2* show upregulation in the adult *Ythdf2* cKO retina compared with control (*Figure 8—figure supplement 1C*). We further found that *Hspa12a* and *Islr2* are downregulated in the retina after AOH operation (*Figure 8—figure supplement 1E*), which is likely caused by upregulation of YTHDF2 in the AOH-treated retina (*Figure 8—figure supplement 1F-H*). We therefore hypothesized that AOH upregulates YTHDF2 which in turn downregulates its targets *Hspa12a* and *Islr2*, thus causing RGC dendrite degeneration and soma loss.

If this is the case, overexpression of *Hspa12a* and *Islr2* might protect RGC dendrite from AOH triggered degeneration. We thus generated AAV harboring overexpression constructs of *Hspa12a* and *Islr2* which were intravitreally injected to wild type retinas. After the AOH induction, the retinas overexpressing *Hspa12a* and *Islr2* maintain significantly more complex RGC dendrite arbor compared with control AAV (*Figure 8E,F*).

These data support a model that AOH upregulates YTHDF2 which in turn downregulates its targets *Hspa12a* and *Islr2*, thus causing RGC dendrite degeneration and eventually resulting in soma loss

2. In figure 7B, YTHDF2 expression in Brn3a+ RGCs was quantified. The figure showed that Brn3a was not detected in some cells. In figure 7C, mRNA level was quantified with an extraction method that did not separate Brn3a+ and Brn3a- cells.

We thank the reviewer for pointing out this. Brn3a is a major RGC marker but does not stain all RGCs. We also used the pan-RGC marker RBPMS. As shown ion *Figure 1A*, all RGCS marked by the pan-RGC marker RBPMS express YTHDF2 while all YTHDF2-expressing cells are RBPMS^+^ RGCs. These data suggest that all RGCs express YTHDF2 and all YTHDF2-expressing cells in retina are RGCs.

3. The authors showed AOH-induced loss of Brn3a+ RGCs in mouse models. Supportive evidence at the phenotype level such as less severe loss of visual acuity in YTHDF2 cKO mice are required.

We thank the reviewer for the suggestion. We have tried to do the OMR assay using the AOH model mice. Unfortunately, the AOH model mice are not eligible for the OMR assay to test their visual acuity. However, we did find that the *Ythdf2* cKO retina with AOH operation maintains significantly higher dendrite complexity compared with the glaucomatous eyes of *Ythdf2^fl/fl^* control mice (*Figure 8A,B*). In addition, the reduction of RGC number in the *Ythdf2* cKO retina with AOH operation is less than control retina with AOH operation (*Figure 8C,D*). These results support that *Ythdf2* cKO protects retina from RGC dendrite degeneration and soma loss caused by AOH.

4. The authors showed knockdown of Kalrn, Strn, and Ubr4 are sufficient to impair dendrite growth. Rescue experiments will further back up their claim.

We thank the reviewer for the suggestion. We have now further performed rescue experiments to examine whether these target mRNAs mediate YTHDF2-regulated RGC dendrite branching. As shown in *Figure 2E-H* and *Figure 3*, cKO of *Ythdf2* led to increased dendrite branching of RGCs both in vitro and in vivo. Transfection of siRNAs against these target mRNAs rescued dendrite branching increases in cultured *Ythdf2* cKO RGCs (*Figure 7C*). We continued to generate and performed intravitreal injection of AAV viral *shKalrn12* and *shUbr4*, which significantly rescued dendrite branching increases of CART^+^ ooDSGCs and SMI-32^+^ αRGCs in *Ythdf2* cKO retina in vivo (*Figure 7D*). These data suggest that these target mRNAs mediate YTHDF2-controlled RGC dendrite branching.

5. The authors suggested that m6A-YTHDF2 axis regulates the stability of Kalrn, Strn, and Ubr4 mRNAs. The authors will need to show these RNAs contain m6A by m6A-IP-qPCR. The effect on Kalrn, Strn, and Ubr4 mRNAs upon m6A writer complex knockdown can also be shown.

We thank the reviewer for the suggestion. We now verified the m^6^A modification of these mRNAs by m6A-IP-qPCR (*Figure 6D*). We also examined whether m^6^A modification regulates the expression levels of these target mRNAs. As shown in *Figure 7—figure supplement 1D*, the mRNA levels of *Kalrn7*, *Kalrn9*, *Kalrn12*, *Strn* and *Ubr4* were dramatically increased after KD of METTL14, supporting that the stability of these target mRNAs is controlled in an m^6^A-dependent manner.

Reviewer #2:This manuscript investigates the role of the m6A reader YTHDF2 during retinal ganglion cell (RGC) dendrite development and branching. The data presented in this study shows that deletion of YTHDF2 in the retina led to increased RGC dendrite branching and improved visual acuity. Based on high throughput analysis, YTHDF2 is suggested to regulate dendrite branching by reducing the stability of kalirin, Strn and Ubr4 mRNAs. Lastly, loss of YTHDF2 is shown to prevent RGC loss in a glaucoma model. The experimental designs were presented clearly, and the results were interpreted with the appropriate statistical tests. However, the phenotype of the Ythdf2 cKO mouse model is not well characterized and there lacks in vivo evidence for the putative targets of YTHDF2. It is also difficult to explain the connection, if any, between the dendritic role of YTHDF2 and its neuroprotective function.1. The authors used RGC cultures to show that knockdown/knockout of YTHDF2 resulted in increased the dendrite branching based on the number of interactions in Sholl analysis. However, to determine the full extent of dendritic defect, it is important to perform detailed analysis of the dendrite structure, including the total dendritic length, number of total segments, maximum branch length, branch order and number of branches.

We thank the reviewer for the suggestions. We have now performed more detailed analysis of the dendrite structure of cultured RGCs after KD of YTHDF2. As shown in *Figure 1—figure supplement 1G,H*, the total dendrite length showed significant increase after YTHDF2 while the length of maximum branch was not changed.

We further focused on the physiological dendrite structures by checking the RGC dendrites in the *Ythdf2* cKO retina in vivo. As shown in *Figure 3—figure supplement 1A-E*, the total dendrite length, number of branches, number of total segments, and number of segments in each branch order of melanopsin^+^ RGCs in the *Ythdf2* cKO retina showed significant increases compared with control retina, while the length of maximum branch was not changed.

2. It is suggested in the manuscript that loss of YTHDF2 in the retina only affects dendrite development of RGCs, but not other cell types. This is based on the simple counting of the number of RPCs at E15 and bipolar and amacrine cells at P15, glossing over the vast complexity of the visual circuitry in the retina. At the very least, the authors should confirm that differentiation of all major retinal neurons and proper formation of the ribbon synapses were unaffected in the cKO mouse model of YTHDF2, which would other confound the interpretation of the RGC and visual acuity phenotype.

Retinal progenitors, amacrine cells, bipolar cells, photoreceptors, or horizontal cells were not affected in *Ythdf2* cKO retina (*Figure 2—figure supplement 1B-L*), suggesting that YTHDF2 is not involved in the generation or development of these cells. This is in line with the low YTHDF2 expression in these cells. The RGC number or density was not affected in the *Ythdf2* cKO retina (*Figure 2C,D*), indicating that *Ythdf2* cKO does not disturb RGC neurogenesis. These data suggest that differentiation of all major retinal neurons was unaffected in the *Ythdf2* cKO mouse model.

In addition, the numbers of the excitatory ribbon synapses marked by the colocalization of Bassoon^+^/PSD-95^+^ in OPL show no difference between *Ythdf2* cKO and control retinas (*Figure 4— figure supplement 1E,F*).

3. The in vivo characterisation of RGC dendrite morphology should be more thorough as previously mentioned, including the arbor shape, size and tiling pattern. The image quality of Melanopsin and SMI-32 staining is low. There are too many cells labelled, making it difficult to examine individual cells/dendrites. Higher resolution images from more sparsely labelled retina would be needed.

We thank the reviewer for the suggestion. We now have repeated and chosen better representative images for melanopsin and SMI-32 staining of RGC dendrites in vivo in the revised manuscript (*Figure 3C,E*). As mentioned earlier, we performed more detailed analysis of the RGC dendrite structures in vivo. As shown in *Figure 3—figure supplement 1A-E*, the total dendrite length, number of branches, number of total segments, and number of segments in each branch order of melanopsin^+^ RGCs in the *Ythdf2* cKO retina showed significant increases compared with control retina, while the length of maximum branch was not changed.

4. The authors provide evidence of increased synaptic density in the IPL using PSD-95 staining in the cKO mice compared to the control litter mates. It is important to investigate if there are any changes to the synaptic density in the OPL. In addition, it is also important to study the number of successful synapse formation by co-staining the tissues with presynaptic markers such as bassoon with PSD-95 and count for co-localisation.

We thank the reviewer for the suggestion. We now used co-staining of the presynaptic marker Bassoon and the postsynaptic marker PSD-95 to count the colocalization puncta of Bassoon^+^/PSD95^+^. We found that the numbers of Bassoon^+^/PSD-95^+^ excitatory synapses in IPL of *Ythdf2* cKO retina are significantly larger than that of control retina (*Figure 4D,E*). As a control, the numbers of the excitatory ribbon synapses marked by the colocalization of Bassoon^+^/PSD-95^+^ in OPL show no difference between *Ythdf2* cKO and control retinas (*Figure 4—figure supplement 1E,F*).

5. The improved visual acuity in Six3Cre-mediated Ythdf2 cKO is solely attributed to the RGC dendrite defects. However, Six3Cre is widely expressed in the ventral telencephalon and ventral anterior hypothalamus, including the SCN and SPZ (PMID: 24767996). It is necessary to examine the retinocollicular projections of the RGC axons.

We thank the reviewer for the suggestion. We checked the targeting of optic nerves to the brain by anterograde labeling with cholera toxin subunit B (CTB) and found no difference of retinogeniculate or retinocollicular projections between *Ythdf2* cKO and control mice (*Figure 5—figure supplement 1F,G*), suggesting the guidance and central targeting of RGC axons are not affected in the *Ythdf2* cKO.

6. The Optomotor response (OMR)-based assay is not the most appropriate test to test the response of ganglion cells since the improved OMR could be attributed to other cell types as previously mentioned. A more targeted approach such as pattern ERG could provide more evidence to their claims.

We tried to find a way to do pattern ERG, but unfortunately we do not have this and could not find reachable resources to do this, either. However, based on the analysis we have done so far, we believe that the improved OMR is most likely attributed to the increased RGC dendrite branching and thicker and denser IPL with more synapses because all other parts and processes of retina are not affected except RGC dendrite in the *Ythdf2* cKO mediated by *Six3-cre:*

(1) As mentioned earlier, the neurogenesis of all major neuron types in retina is not affected by *Ythdf2* cKO in retina (*Figure 2C,D*; *Figure 2—figure supplement 1B-L*).

(2) The increases of RGC dendrite branching in the *Ythdf2* cKO retina were validated both in vitro and in vivo (*Figure 2E-H*; *Figure 3*).

(3)IPL thickness significantly increased in the *Ythdf2* cKO retina (*Figure 4A,B*). As a control, the thicknesses of other retinal layers (ONL, OPL, INL, GCL) showed no difference between the *Ythdf2* cKO and control mice (*Figure 4—figure supplement 1A-D*).

The numbers of Bassoon^+^/PSD-95^+^ excitatory synapses in IPL of *Ythdf2* cKO retina are significantly larger than that of control retina (*Figure 4D,E*). As a control, the numbers of the excitatory ribbon synapses marked by the colocalization of Bassoon^+^/PSD-95^+^ in OPL show no difference between *Ythdf2* cKO and control retinas (*Figure 4—figure supplement 1E,F*).

7. The quality of the RIP-seq analysis is not clear. Is there an isotype antibody control? Are there biological replicates? What is the statistical significance of the top hits? There are similar concerns regarding the MS analysis and whether there is significant overlap between the target lists generated from RIP-Seq and MS studies.

We are sorry that the related information was not clear or missing. For the RIP experiment: Yes, we used a control IgG and performed two biological replicates. To determine which gene is enriched, we computed the FPKM from RIP elute to input and any fold change greater than 2 (p value less than 0.05) was considered enriched. For MS analysis, we performed three biological replicates, and proteins with fold changes greater than 1.3 and p values less than 0.05 were considered to be regulated by YTHDF2 KD with statistical significance. By overlapping the two gene lists screened from anti-YTHDF2 RIP-Seq (*Supplementary file 1*) and YTHDF2 KD/MS_upregulation (*Supplementary file 2*), we identified a group of potential YTHDF2 target mRNAs in RGCs (*Supplementary file 3*).

As a summary, we now have added these information to the Materials and methods, and a new *Supplementary file 3* in the revised manuscript.

8. The authors provide evidence in the RGC culture that stabilisation of kalirin, Strn and Ubr4 mRNAs could contribute to the increase in dendrite branching in Ythdf2 knockout. The regulation of these genes by Ythdf2 need to be validated in Ythdf2 cKO retina at the level of mRNA by qRT-PCR and at the level of protein by western blot. Functionally, the authors need to investigate if knockdown or knockout of these genes in retina indeed affect dendrite branching or even rescue Ythdf2 cKO phenotype.

As shown in *Figure 7—figure supplement 1B*, the mRNA levels of *Kalrn7*, *Kalrn9*, *Kalrn12*, *Strn* and *Ubr4* were dramatically increased after KD of YTHDF2 by qRT-PCR. We further verified this by directly measuring the stability of these target mRNAs. As shown in *Figure 7A*, all the target mRNAs showed significantly increased stability in the *Ythdf2* cKO retina compared with controls. These results suggest that YTHDF2 controls the levels of its m^6^A-modifed target mRNAs by decreasing their stability.

MS analysis after YTHDF2 KD has shown that the protein levels of these target mRNAs were upregulated (*Supplementary file 2*). For validation, we performed IF using antibodies against Strn and Ubr4 (unfortunately we could not find any convincing Abs against Kalrns after many attempts) and detected specific signals in the IPL which were increased in *Ythdf2* cKO retina compared with control retina (*Figure 7—figure supplement 1A*), suggesting that Strn and Ubr4 are upregulated in Ythdf2 cKO retina at the level of protein.

We continued to explore the functions of these YTHDF2 target mRNAs in RGC dendrite development. We first generated siRNAs against these transcripts (*Figure 7—figure supplement 1E*). We then checked the effects on RGC dendrite branching after KD of these target mRNAs by siRNAs in cultured RGCs. As shown in *Figure 7B*, KD of *Kalrn7*, *Kalrn9*, *Kalrn12*, *Strn* or *Ubr4* led to significant decreases of RGC dendrite branching. We further examined whether these target mRNAs mediate YTHDF2regulated RGC dendrite branching. As shown in *Figure 2E-H*, and *Figure 3*, cKO of *Ythdf2* led to increased dendrite branching of RGCs both in vitro and in vivo. Transfection of siRNAs against these target mRNAs rescued dendrite branching increases in cultured *Ythdf2* cKO RGCs (*Figure 7C*). We continued to perform intravitreal injection of AAV viral *shKalrn12* and *shUbr4*, which significantly rescued dendrite branching increases of CART^+^ ooDSGCs and SMI-32^+^ αRGCs in *Ythdf2* cKO retina in vivo (*Figure 7D*). These data support that knockdown of these genes indeed reduces the RGC dendrite branching, and can rescue the *Ythdf2* cKO phenotype both in vitro and in vivo.

9. Finally, there is not clear connection between the dendrite development phenotype and the neuroprotective role of YTHDF2. The authors need to clarify the rationale behind these experiments.

We thank the reviewer for pointing out this. To answer this question, we now have expanded the scope of our study in the revised manuscript.

The pathological changes in RGC dendrites precede axon degeneration or soma loss in the glaucomatous eyes of different model animals (Morgan et al., 2006; Shou et al., 2003; Weber et al., 1998). Our findings that *Ythdf2* cKO in retina promotes RGC dendrite branching during development inspired us to wonder whether YTHDF2 also regulates RGC dendrite maintenance in the acute glaucoma model caused by acute ocular hypertension (AOH).

We found that two target mRNAs *Hspa12a* and *Islr2* show upregulation in the adult *Ythdf2* cKO retina compared with control (*Figure 8—figure supplement 1C*). We further found that *Hspa12a* and *Islr2* are downregulated in the retina after AOH operation (*Figure 8—figure supplement 1E*), which is likely caused by upregulation of YTHDF2 in the AOH-treated retina (*Figure 8—figure supplement 1F-H*). We therefore hypothesized that AOH upregulates YTHDF2 which in turn downregulates its targets *Hspa12a* and *Islr2*, thus causing RGC dendrite degeneration and soma loss. If this is the case, overexpression of *Hspa12a* and *Islr2* might protect RGC dendrite from AOH-triggered degeneration. We thus generated AAV harboring overexpression constructs of *Hspa12a* and *Islr2* which were intravitreally injected to wild type retinas. After the AOH induction, the retinas overexpressing *Hspa12a* and *Islr2* maintain significantly more complex RGC dendrite arbor compared with control AAV (*Figure 8E,F*). These data support a model that AOH upregulates YTHDF2 which in turn downregulates its targets *Hspa12a* and *Islr2*, thus causing RGC dendrite degeneration and eventually resulting in soma loss.

The mechanistic link between the dendrite development phenotype and the neuroprotective role of YTHDF2 is that YTHDF2 has two phases of function to control RGC dendrite development first and then maintenance through regulating two sets of target mRNAs. In early postnatal stages, the target mRNAs *Kalrn7*, *Kalrn9*, *Kalrn12*, *Strn* and *Ubr4* mediate YTHDF2 functions to regulate RGC dendrite development. In adult mice, another set of target mRNAs *Hspa12a* and *Islr2* mediate YTHDF2 function to regulate RGC dendrite maintenance. We now added more discussion and explanation on the connection of the two phases of YTHDF2 function in the revised manuscript.

Reviewer #3:In this study, the authors propose a role for the m6A reader YTHDF2 in regulating the dendritic arbor size of retinal ganglion cells (RGCs), and they extrapolate this to address issues in glaucoma and overall retina IPL morphology. They find that retina-specific loss of Ythdf2 leads to the expansion of RGC dendritic arbors in the horizontal plane and also a widening of the inner plexiform layer (IPL), with Ythdf2 cKOs showing a modest increase in visual acuity in an optomotor assay. They identify a number of factors downstream of YTHDF2 not previously known to be involved in retinal dendritic development, and they propose a role for Ythdf2 in a glaucoma model, providing a foundation for future studies into their functions in these contexts.Although these data are sufficient to establish a role for Ythdf2 in RGC dendritic development and other more general aspects of retina morphology, many of the authors' specific claims are weakened by issues with the experiments, their analyses, and by weak effect sizes. Even if the points raised below were addressed, it remains unclear from this study how Ythdf2-mediated mechanisms are integrated into programs of dendritic development and RGC survival, diminishing the impact of this work. This study would be strengthened by, for example, identifying a developmental program that modifies the target mRNAs found in this study to promote recognition by YTHDF2, and also by providing better understanding of how the products produced by these mRNAs actually regulate dendrite morphology and synapse formation. Overall, while there are a number of promising findings in this study, the work required to address weaknesses falls well outside of the limits usually recommended by eLife, and it is suited for publication in a more specialized journal following attention to many of the points raised below.1. The cell-type specific tracing in Figure 3 is difficult to assess due to the extremely high background in the melanopsin images and unclear signal in the SMI-32 images. This analysis would be improved by sparse viral labeling, with a fluorophore provided by the virus used for tracing and the melanopsin & SMI-32 antibodies used solely for establishing RGC subtype identity. Further, the authors emphasize that the defects they observe here are in the plane of retinal ganglion cell (RGC) dendritic arbors, but this is an over-interpretation of these data. One wonders why the authors did not examine in cross section the lamination within the IPL of the melanopsin-positive ipRGCs, since this would provide, at least for this RGC subclass, a direct assessment of this issue. Finally, the Thy-1-O line GFP distribution is quite general, and it really is not the correct line to use if one wants to investigate defects in lamination.

We thank the reviewer for pointing out this. To answer this question, we now have expanded the scope of our study in the revised manuscript.

The pathological changes in RGC dendrites precede axon degeneration or soma loss in the glaucomatous eyes of different model animals (Morgan et al., 2006; Shou et al., 2003; Weber et al., 1998). Our findings that *Ythdf2* cKO in retina promotes RGC dendrite branching during development inspired us to wonder whether YTHDF2 also regulates RGC dendrite maintenance in the acute glaucoma model caused by acute ocular hypertension (AOH).

We found that two target mRNAs *Hspa12a* and *Islr2* show upregulation in the adult *Ythdf2* cKO retina compared with control (*Figure 8—figure supplement 1C*). We further found that *Hspa12a* and *Islr2* are downregulated in the retina after AOH operation (*Figure 8—figure supplement 1E*), which is likely caused by upregulation of YTHDF2 in the AOH-treated retina (*Figure 8—figure supplement 1F-H*). We therefore hypothesized that AOH upregulates YTHDF2 which in turn downregulates its targets *Hspa12a* and *Islr2*, thus causing RGC dendrite degeneration and soma loss. If this is the case, overexpression of *Hspa12a* and *Islr2* might protect RGC dendrite from AOH-triggered degeneration. We thus generated AAV harboring overexpression constructs of *Hspa12a* and *Islr2* which were intravitreally injected to wild type retinas. After the AOH induction, the retinas overexpressing *Hspa12a* and *Islr2* maintain significantly more complex RGC dendrite arbor compared with control AAV (*Figure 8E,F*). These data support a model that AOH upregulates YTHDF2 which in turn downregulates its targets *Hspa12a* and *Islr2*, thus causing RGC dendrite degeneration and eventually resulting in soma loss.

The mechanistic link between the dendrite development phenotype and the neuroprotective role of YTHDF2 is that YTHDF2 has two phases of function to control RGC dendrite development first and then maintenance through regulating two sets of target mRNAs. In early postnatal stages, the target mRNAs *Kalrn7*, *Kalrn9*, *Kalrn12*, *Strn* and *Ubr4* mediate YTHDF2 functions to regulate RGC dendrite development. In adult mice, another set of target mRNAs *Hspa12a* and *Islr2* mediate YTHDF2 function to regulate RGC dendrite maintenance. We now added more discussion and explanation on the connection of the two phases of YTHDF2 function in the revised manuscript.

2. The authors repeatedly describe increased dendritic and synaptic density within the IPL in Ythdf2 cKOs, yet they only present absolute signal intensities to support these claims; without normalizing signal intensity to IPL width, it is impossible to determine whether changes in intensity are indeed due to increased density or from the IPL becoming enlarged with a similar density of dendrites and synapses. Further, they claim in the text that VGlut2 puncta numbers are changed in these mutants, however the analysis presented is only for general signal intensity. If claims are to be made regarding changes in synaptic specializations, higher resolution assessments of individual puncta using pre- and postsynaptic markers is required.

We are sorry that the information about increased dendritic and synaptic density within the IPL in *Ythdf2* cKO was not clear. The MAP2 IF intensity shown in *Figure 4C* was already normalized to the IPL area. Considering the increased IPL thickness, the absolute signal intensities (MAP2 IF intensity per area × IPL area) would be even larger in the *Ythdf2* cKO compared with the control. We have now made this (MAP2 IF intensity per area in IPL) clearer in the revised manuscript.

We thank the reviewer for the suggestion on quantifying synaptic puncta. We now used co-staining of the presynaptic marker Bassoon and the postsynaptic marker PSD-95 to count the colocalization puncta of Bassoon^+^/PSD-95^+^. We found that the numbers of Bassoon^+^/PSD-95^+^ excitatory synapses in IPL of *Ythdf2* cKO retina are significantly larger than that of control retina (*Figure 4D,E*).

3. While Ythdf2 cKOs do show a statistically significant improvement in visual acuity (although for these male data it is unclear whether the control data distribution is sufficiently normal to use a t-test), the effect size is small enough to call into question how meaningful the improvement is. Further, what is the actual relationship between this modest improvement in optomotor response (OMR) and the morphological phenotypes observed here, in particular since they are not connected here to the classes of direction-selective RGCs one would expect might affect OMR responses.

The difference of 0.02 cycle/degree between young *Ythdf2* cKO and control is small yet significant in both male and female mice in our study, and this difference is comparable with other studies in the field (Dietrich et al., 2019; Kang et al., 2013; Thangthaeng et al., 2017). Considering the fact that young wild-type (control) mice already have good visual acuity, this 0.02 cycle/degree increase is quite significant.

We thank the reviewer for suggesting checking direction-selective RGCs to enhance the relationship between the improvement in optomotor response (OMR) and the morphological phenotypes. The ON-OFF directionally selective RGCs (ooDSGCs) respond preferentially to movement in particular directions. Expression of CART (cocaine- and amphetamine-regulated transcript), a neuropeptide, distinguishes ooDSGCs from other RGCs (Kay et al., 2011). We now found that the dendrite branching of ooDSGCs marked by CART/Brn3a co-staining in *Ythdf2* cKO retinal cultures increased compared with control in vitro (*Figure 2G,H*). Intravitreal injection of an AAV reporter expressing ZsGreen visualized the dendrite morphology of ooDSGCs marked by CART immunostaining in vivo (*Figure 3A*). We found that ooDSGCs showed dramatically increased dendrite branching in *Ythdf2* cKO retina compared with control retina (*Figure 3A,B*). Thus, *Ythdf2* cKO increased the dendrite branching of ooDSGCs both in vitro and in vivo, along with other data in this study supporting the improved optomotor response in the *Ythdf2* cKO mice.

4. Much clearer co-localization between YTHDF2 and RGC markers is needed, both in vivo and in vitro, to clearly demonstrate association with RGCs and not displaced amacrine cells or other retinal cell types.

We now used the pan-RGC marker RBPMS. As shown ion *Figure 1A*, all RGCS marked by the pan-RGC marker RBPMS express high YTHDF2 while all YTHDF2-expressing cells are RBPMS^+^ RGCs. These data demonstrate the co-localization between YTHDF2 and the pan-RGC marker.

5. The several mRNAs and their produces shown to be under YTHDF2 control do appear to play some role in regulating general RGC dendritic morphology, though of course from these data we do not know whether or not there is any RGC-subtype specificity. However, as presented, we really do not learn much at all about their mechanisms of action. At the very least, do they act in the same, or different, pathways….i.e. have any attempts been made to knock down more than one at a time and ask about independent pathway affiliations?

We thank the reviewer for the suggestion. We now have prepared a cocktail of siRNAs against these target mRNAs and performed the KD assay to check the effect on RGC dendrite morphology. As shown in *Figure 7—figure supplement 1F*, the *siCocktail* further significantly reduced the RGC dendrite branching compared with each individual siRNA. These results suggest that these targets may work in different pathways to regulate the RGC dendrite morphology. We agree that the exploration of their working mechanisms is an important future direction.

6. It is not clear how to interpret the Ythdf2 effects in the acute ocular hypertension (AOH) model. Is this related to the downstream mRNAs, or some other more general effect of loss of this m6 reader? And how is this related to RGC survival at a mechanistic level?

We are sorry that this was not clear in the original manuscript. To answer this question, we now have expanded the scope of our study in the revised manuscript. We found that two YTHDF2 target mRNAs *Hspa12a* and *Islr2* show upregulation in the adult *Ythdf2* cKO retina compared with control (*Figure 8—figure supplement 1C*). We further found that *Hspa12a* and *Islr2* are downregulated in the wildtype retina after AOH operation (*Figure 8—figure supplement 1E*), which is likely caused by upregulation of YTHDF2 in the AOH-treated retina (*Figure 8—figure supplement 1F-H*). We therefore hypothesized that AOH upregulates YTHDF2 which in turn downregulates its targets *Hspa12a* and *Islr2*, thus causing RGC dendrite degeneration. If this is the case, overexpression of *Hspa12a* and *Islr2* might protect RGC dendrite from AOH-triggered degeneration. We thus generated AAV harboring overexpression constructs of *Hspa12a* and *Islr2* which were intravitreally injected to wild type retinas. After the AOH induction, the retinas overexpressing *Hspa12a* and *Islr2* maintain significantly more complex RGC dendrite arbor compared with control AAV (*Figure 8E,F*).

AOH has been shown to cause pathological changes in RGC dendrites before axon degeneration or soma loss is detected in different model animals (Morgan et al., 2006; Shou et al., 2003; Weber et al., 1998), suggesting that RGC soma loss might be following dendrite degeneration. Thus, these findings support a model that AOH upregulates YTHDF2 which in turn downregulates its targets *Hspa12a* and *Islr2*, thus causing RGC dendrite degeneration and eventually resulting in soma loss.

Summary

As a summary, we have now substantially expanded and improved our study to address all the concerns raised by the reviewers and summarized by the editor:

1. We have now done a better phenotyping of the *Ythdf2* cKO mouse model by more precisely measuring the layer dimensions (*Figure 4—figure supplement 1A-D*), the synapse number (*Figure 4D,E*), the dendritic extension (*Figure 1—figure supplement 1G,H*; *Figure 3—figure supplement 1A-E*); checking cells other than RGCs in retina (*Figure 2—figure supplement 1B-L*); confirming colocalization of YTHDF2 and RGC markers (*Figure 1A*). We also explained the connection between the dendrite development phenotype and the neuroprotective role of YTHDF2 (please see our response to Reviewer #2’s major comment 9 for details).

2. We have now provided in vivo evidence to show that the YTHDF2 target mRNAs *Kalirin*, *Strn* and *Ubr4* indeed contribute to the increase in dendrite branching in the *Ythdf2* knockout. We generated and performed intravitreal injection of AAV viral *shKalrn12* and *shUbr4*, which significantly rescued dendrite branching increases of CART^+^ ooDSGCs and SMI-32^+^ αRGCs in *Ythdf2* cKO retina in vivo (*Figure 7D*). These data suggest that these target mRNAs mediate YTHDF2-controlled RGC dendrite branching. As for the mechanism of loss of Brn3a^+^ RGCs mediated by YTHDF2 upregulation in AOHtreated retina, please see our response to Reviewer #1’s comment 1 for details.

3. We have now included the analysis of the direction-selective RGCs to enhance the relationship between the improvement in optomotor response (OMR) and the morphological phenotypes (please see our response to Reviewer #3’s comment 3 for details).

4. By expanding the scope of our study, we have now more clearly interpreted the YTHDF2 effects in the acute ocular hypertension (AOH) model, which are mediated by the two YTHDF2 target mRNAs *Hspa12a* and *Islr2* (please see our response to Reviewer #3’s comment 6 for details).

5. We used a model to explain the relationship among glaucoma, YTHDF2, and RGC survival: AOH upregulates YTHDF2 which in turn downregulates its targets *Hspa12a* and *Islr2*, thus causing RGC dendrite degeneration and eventually resulting in soma loss.

We think that the revised manuscript is substantially improved with these revisions and new data.

We would like to thank the reviewers for their comments and suggestions.

References

Dietrich, M., Hecker, C., Hilla, A., Cruz-Herranz, A., Hartung, H.P., Fischer, D., Green, A., and Albrecht, P. (2019). Using Optical Coherence Tomography and Optokinetic Response As Structural and Functional Visual System Readouts in Mice and Rats. J Vis Exp.

Feng, G., Mellor, R.H., Bernstein, M., Keller-Peck, C., Nguyen, Q.T., Wallace, M., Nerbonne, J.M., Lichtman, J.W., and Sanes, J.R. (2000). Imaging neuronal subsets in transgenic mice expressing multiple spectral variants of GFP. Neuron 28, 41-51.

Kang, E., Durand, S., LeBlanc, J.J., Hensch, T.K., Chen, C., and Fagiolini, M. (2013). Visual acuity development and plasticity in the absence of sensory experience. J Neurosci 33, 17789-17796.

Kay, J.N., De la Huerta, I., Kim, I.J., Zhang, Y., Yamagata, M., Chu, M.W., Meister, M., and Sanes, J.R. (2011). Retinal ganglion cells with distinct directional preferences differ in molecular identity, structure, and central projections. J Neurosci 31, 7753-7762.

Morgan, J.E., Datta, A.V., Erichsen, J.T., Albon, J., and Boulton, M.E. (2006). Retinal ganglion cell remodelling in experimental glaucoma. Adv Exp Med Biol 572, 397-402.

Shou, T., Liu, J., Wang, W., Zhou, Y., and Zhao, K. (2003). Differential dendritic shrinkage of α and β retinal ganglion cells in cats with chronic glaucoma. Invest Ophthalmol Vis Sci 44, 3005-3010.

Thangthaeng, N., Rutledge, M., Wong, J.M., Vann, P.H., Forster, M.J., and Sumien, N. (2017). Metformin Impairs Spatial Memory and Visual Acuity in Old Male Mice. Aging Dis 8, 17-30.

Weber, A.J., Kaufman, P.L., and Hubbard, W.C. (1998). Morphology of single ganglion cells in the glaucomatous primate retina. Invest Ophthalmol Vis Sci 39, 2304-2320.

[Editors’ note: what follows is the authors’ response to the second round of review.]

The manuscript has been improved but there are some remaining issues that need to be addressed, as outlined below and in the appended reviews:Reviewer 2 (previous Reviewer 1) was satisfied with the revisions of this resubmission as was Reviewer 3 (previous Reviewer 2 although the latter reviewer adds that "…we still are far from a mechanistic understanding of how these gene products regulate RGC dendritic arborization, and whether or not there is any specificity of these effects with regard to RGC subtypes."Reviewer 1 (previous Reviewer 2) continues to have concerns: He summarized in the Public Review (this will not be made public as it is a resubmission) that "the phenotype of the Ythdf2 cKO mouse model is not well characterized and there lacks in vivo evidence for the putative targets of YTHDF2".

We thank the reviewer for pointing out these important future directions. We have now added related discussions in the revised manuscript.

Two items he previously requested were not addressed:(1) To perform either qPCR or western blots of Ythdf2 cKO retina to confirm the regulation of Kalirin, Strn and Ubr4 by Ythdf2.

We have now carried out qPCR for *Kalirin*, *Strn*, *Ubr4* using the *Ythdf2* cKO and control retina, confirming that *Kalirin*, *Strn*, and *Ubr4* mRNA levels were upregulated in the *Ythdf2* cKO retina compared with control (*Figure 7—figure supplement 1C*).

2) To show that overexpression of Hspa12a and Islr2 alleviates loss of RGCs in the AOH model (See point #8 of previous Reviewer 2’s review). We ask that you address these two items that were raised in the first review, to the best of your ability, given time constraints. (Thus, these are not experiments newly requested by the reviewer(s).

In the original manuscript, we have shown that the retinas overexpressing Hspa12a and Islr2 maintain significantly more complex RGC dendrite arbor compared with control AAV after the AOH induction (*Figure 8E,F*). We now have repeated this experiment to further check loss of RGCs. As shown in the revised *Figure 8G*, retinas overexpressing Hspa12a and Islr2 maintain significantly more Brn3a^+^ RGCs than control AAV after the AOH induction, suggesting that overexpression of Hspa12a and Islr2 alleviates loss of RGCs in the AOH model.

Current Reviewer 1 has also raised additional points not in the first review, that could be answered mostly textually:1. Your stated exclusivity of YTHDF2 expression in RGCs, given that there is apparent staining in all cell layers in Suppl Figure 1B. He cited very "old"scRNAseq data indicating that YTHDF2 is found in all retinal cell types.

YTHDF2 is highly expressed in RGCs (*Figure 1A*; *Figure 1—figure supplement 1B*). Conversely, the expression of YTHDF2 in other layers of the retina is much lower (*Figure 1A*; *Figure 1—figure supplement 1B*). Though the expression of YTHDF2 could be detected by scRNAseq in other retinal cell types in a previous report as the reviewer mentioned, their YTHDF2 expression levels are much lower than RGCs. More importantly, consistent with the minimal or no expression of YTHDF2 in other retinal layers/cells, all other retinal layers/cells except IPL are not affected in the *Ythdf2* cKO retina (*Figure 2—figure supplement 1; Figure 2—figure supplement 2*).

2. To verify that YTHDF2 is neuronal: determining YTHDF2 expression in Muller glia and astrocytes.

GFAP is reliable marker for astrocytes in the mammalian retina (Vecino et al., 2016). As shown in *Figure 1—figure supplement 1C,D*, YTHDF2 expression could not be detected in these cells at P20, suggesting that YTHDF2 expression in retina is neuronal (RGCs).

3. Addressing the utility of the Six3-Cre, by determining whether there is mosaic expression of Cre in the peripheral retina.

As reported previously, there is minimal mosaic absence of expression of Six3-Cre in the peripheral retina as shown in *Figure 5—figure supplement 1A*. The fact that the layer of Brn3a^+^ RGCs become thinner and thinner toward the edge of peripheral retina (Quina et al., 2005) makes the trace residual expression of YTHDF2 in the peripheral retina of *Ythdf2* cKO mice negligible (Author response image 1). Indeed, YTHDF2 depletion in the Six3-Cre-mediated *Ythdf2* cKO retina is robust and efficient (*Figure 2B*).

**Author response image 1. sa2fig1:** The trace residual expression of YTHDF2 in the peripheral retina of *Ythdf2* cKO mice. The representative confocal images of YTHDF2 IF in E15.5 *Ythdf2* cKO retina shows the trace residual expression of YTHDF2 in the peripheral retina of *Ythdf2* cKO mice. Scale bar, 100 µm.

Reviewer #1:The experimental designs were presented clearly and the results were interpreted with the appropriate statistical tests. However, the phenotype of the Ythdf2 cKO mouse model is not well characterized and there lacks in vivo evidence for the putative targets of YTHDF2. It is also difficult to explain the connection, if any, between the dendritic role of YTHDF2 and its neuroprotective function. Most importantly, whether Ythdf2 acts cell autonomously in the RGCs remains to be resolved.1. The premise of this study is that Ythdf2 is cell autonomously required in RGCs for dendritic development. In the rebuttal letter, the authors even went as far as claiming that "all RGCs express YTHDF2 and all YTHDF2-expressing cells in retina are RGCs". However, this statement is belied by the strong YTHDF2 staining outside the RGC layers in Figure 1—figure supplement 1B. The YTHDF2 expression in non-RGC cells can also be seen in Figure 1A, although for unknown reason the staining intensity was weaker. In fact, numerous RNAseq studies have now thoroughly catalogued gene expression patterns in the retina throughout development and YTHDF2 expression can be found among all retinal cell types even from the earliest scRNAseq data (Macosko et al., 2015, Cell 161, 1202). The ubiquitous expression of YTHDF2 in the retina poses serious concerns regarding the usefulness of Six3-Cre mediated Ythdf2 cKO model and the validity of presumed Ythdf2 downstream targets in RGCs.

This has been addressed in the response to reviewer 1’s point 1 summarized by the editor.

2. The revised manuscript presents data to show that retinal progenitors, amacrine cells, bipolar cells, photoreceptors, or horizontal cells were not affected in Ythdf2 cKO retina. It is also important for the authors to examine Müller glia and astrocyte, especially the latter which is known to have a strong influence on RGC activity and health.

We now have checked Lhx2^+^ Müller glia and GFAP^+^ astrocytes in P20 *Ythdf2* cKO retina. Neither of these two cell populations was affected in the *Ythdf2* cKO retina (*Figure 2—figure supplement 2*), which is not surprising since these cells don’t express YTHDF2 (*Figure 1—figure supplement 1C,D*)

3. The potential targets of YTHDF2 were mostly characterized in the RGC culture, which is not a good model for studying dendritic development because RGCs will evitable die in vitro. As previously requested by this reviewer, the authors need to perform either qPCR or western blos of Ythdf2 cKO retina to confirm the regulation of Kalirin, Strn and Ubr4 by Ythdf2. They also need to show whether siRNA knockdown or CRISPR knockout of these genes rescue the Ythdf2 cKO phenotype in vivo.

This has been addressed in the response to the editor’s emphasized item (1). For the rescue experiments in vivo, these have been done (*Figure 7D*).

4. The authors propose that the better maintenance of RGC dendrite arborization in Ythdf2 cKO retina make it resistant to RGC death in AOH model and this is mediated by Hspa12a and Islr2 genes. To support this claim, the authors need to show that overexpression of Hspa12a and Islr2 alleviates loss of RGCs in the AOH model.

This has been addressed in the response to the editor’s emphasized item (2).

Reviewer #2:The authors have responded to my comments and suggestions with several new experiments and some additional explanations. Though the effects of YTHDF2 described here are pleiotropic across RGC cell types, these defects in dendritic arborization are better described in this revised manuscript. In addition, further attention to the mRNAs regulated by YTHDF2 is given, and though we still are far from a mechanistic understanding of how these gene products regulate RGC dendritic arborization, and whether or not there is any specificity of these effects with regard to RGC subtypes, I find this revision to be a substantial improvement over the initial submission.

We thank the reviewer for pointing out these important future directions. We have now added related discussions in the revised manuscript.

References

de Melo, J., Zibetti, C., Clark, B.S., Hwang, W., Miranda-Angulo, A.L., Qian, J., and Blackshaw, S. (2016). Lhx2 Is an Essential Factor for Retinal Gliogenesis and Notch Signaling. J Neurosci *36*, 2391-2405. 10.1523/jneurosci.3145-15.2016.

Quina, L.A., Pak, W., Lanier, J., Banwait, P., Gratwick, K., Liu, Y., Velasquez, T., O'Leary, D.D., Goulding, M., and Turner, E.E. (2005). Brn3a-expressing retinal ganglion cells project specifically to thalamocortical and collicular visual pathways. J Neurosci *25*, 11595-11604. 10.1523/jneurosci.2837-05.2005.

Vecino, E., Rodriguez, F.D., Ruzafa, N., Pereiro, X., and Sharma, S.C. (2016). Glia-neuron interactions in the mammalian retina. Prog Retin Eye Res *51*, 1-40. 10.1016/j.preteyeres.2015.06.003.